# Optimizing Water and Nitrogen Management Strategies to Unlock the Production Potential for Onion in the Hexi Corridor of China: Insights from Economic Analysis

**DOI:** 10.3390/plants15010006

**Published:** 2025-12-19

**Authors:** Xiaofan Pan, Haoliang Deng, Guang Li, Qinli Wang, Rang Xiao, Wenbo He, Wei Pan

**Affiliations:** 1College of Civil Engineering, Hexi University, Zhangye 734000, China; panxiaofan2023@163.com (X.P.); denghaoliang521@163.com (H.D.); xiaorang999@163.com (R.X.); 2Gansu Provincial Engineering Research Center for the Resource Utilization of Edible Fungi and Fungi Bran, Hexi University, Zhangye 734000, China; wangqinli66@163.com; 3College of Agriculture and Ecological Engineering, Hexi University, Zhangye 734000, China; 4College of Water Conservancy and Hydropower Engineering, Gansu Agricultural University, Lanzhou 730070, China; hewenbo2025@163.com (W.H.); panwei2025666@163.com (W.P.)

**Keywords:** water and nitrogen regulation, onion, economic benefits, quality, yield, water and nitrogen use efficiency

## Abstract

Water and nitrogen are the key factors restricting the productivity improvement of onion in the Hexi Oasis. Unreasonable water and fertilizer management not only increases input costs, but also causes environmental pollution of farmland soil, thereby affecting the sustainable development of agriculture. To explore the effects of the water–nitrogen interaction and optimized combination schemes on onion yield, water–nitrogen use efficiency, and economic benefits under mulched drip irrigation in the Hexi Oasis, a four-year (2020–2023) water–nitrogen coupling regulation experiment was conducted at the Yimin Irrigation Experimental Station in Minle County, Hexi Corridor. The onion was used as the test crop and three irrigation levels were established, based on reference crop evapotranspiration (ET_c_): low water (W1, 70% ET_c_), medium water (W2, 85% ET_c_), and sufficient water (W3, 100% ET_c_), as well as high nitrogen N3 (330 kg·ha^−1^), medium nitrogen N2 (264 kg·ha^−1^), and low nitrogen N1 (198 kg·ha^−1^). Meanwhile, no nitrogen application N0 (0 kg·ha^−1^) was set as the control at three irrigation levels. This study analyzed the effects of different water and nitrogen supply conditions on onion quality, yield, water–nitrogen use efficiency, and economic benefits. A water–nitrogen economic benefit coupling model was established to optimize water–nitrogen combination schemes targeting different economic objectives. The results revealed that medium-to-high water–nitrogen combinations were beneficial for improving onion quality, while excessive irrigation and nitrogen application inhibited bulb quality accumulation. Both yield and economic benefits increased with the increasing amount of irrigation, whereas excessive nitrogen application showed a diminishing yield-increasing effect, simultaneously increasing farm input costs and ultimately reducing the economic benefits. In the four-year experiment, the N3W3 treatment in 2020 achieved the highest yield, economic benefits, and net profit, reaching 136.93 t·ha^−1^, 20,376.3 USD·ha^−1^, and 14,320.8 USD·ha^−1^, respectively, with no significant difference from the N2W3 treatment. From 2021 to 2023, the N2W3 treatment achieved the highest yield, economic benefits, and net profit, averaging 130.87 t·ha^−1^, 28,449.5 USD·ha^−1^, and 21,881.5 USD·ha^−1^, respectively. Lower irrigation and nitrogen application rates mutually restricted the water and nitrogen utilization, resulting in low water use efficiency, irrigation water use efficiency, nitrogen partial factor productivity, and nitrogen agronomic use efficiency. The relationship between the irrigation amount, nitrogen application rate, and the economic benefits of onion fits a bivariate quadratic regression model. This model predicts that onion’s economic benefits are highly correlated with the actual economic benefits, with analysis revealing a parabolic trend in economic benefits as water and nitrogen inputs increase. By optimizing the model, it was determined that when the irrigation amount reached 100%, the ET_c_ and nitrogen application rate was 264 kg·ha^−1^, and the economic benefits were close to the target range of 27,000–29,000 USD·ha^−1^; this can be used as the optimal water and nitrogen management model and technical reference for onion in the Hexi Oasis irrigation area, which can not only ensure high yield and quality but also improve the use efficiency of water and nitrogen.

## 1. Introduction

The Hexi Corridor, as a major production area for grain and cash crops in the arid northwest region of China, holds significant strategic importance in ensuring regional food security and maintaining ecological barriers [1]. The area is characterized by a pronounced arid climate, with the annual precipitation averaging less than 200 mm and evaporation exceeding 2000 mm. Water resource constraints have thus become a critical limiting factor for sustainable agricultural development [2]. Zhangye City, as a typical oasis agricultural area in the Hexi Corridor, had a vegetable planting area of 49,000 hectares in 2024. Among these crops, onion, with its drought-resistant characteristics, has become a dominant cash crop in the region, with a stable planting scale of over 5000 hectares and an annual output of up to 500,000 tons. However, under intensive planting practices, the imbalance in water and nitrogen resource management has become increasingly prominent [3,4]. Research shows that traditional onion cultivation irrigation quotas are generally high, and water resource use efficiency is significantly lower than the national benchmark level [5]. This inefficient management model not only causes ineffective waste of water resources but also exacerbates the negative impact of extreme climate change on yield stability [6], posing a serious threat to regional agricultural ecological security.

Water regulation is a key driving factor in agricultural production within arid regions, and its optimal management has multifaceted regulatory effects on crops’ physiological and ecological processes [7]. Moderate water stress can induce adaptive responses in crops, prioritizing the allocation of photosynthetic products to the root system, promoting vertical root development, and enhancing soil water and nutrient capture capacity [8,9]. It can also facilitate stomatal regulation, thereby improving water use efficiency [8,9]. Furthermore, by regulating secondary metabolic pathways in crops, it promotes the synthesis and accumulation of soluble sugars, vitamins, and phenolic compounds, thereby enhancing crops’ nutritional quality and flavor characteristics [10]. However, excessive water stress can cause damage to the photosynthetic system and imbalance in metabolic homeostasis, subsequently reducing the commercial value of agricultural products [11]. Scientific irrigation not only improves the water use efficiency but also enhances the bioavailability of nutrients such as nitrogen by regulating the root architecture and improving the rhizosphere microenvironment, thereby achieving water–nitrogen synergistic effects [12,13]. However, the current research is mostly limited to single-factor regulation of water thresholds, and there remains a significant gap in the systematic analysis of water–nitrogen interaction mechanisms.

Nitrogen fertilizer management exerts a dual regulatory effect on crop yield formation. Appropriate nitrogen application promotes leaf expansion, enhances Rubisco enzyme activity [14], and increases the soluble protein content in functional leaves [15], laying the foundation for high yields. However, excessive nitrogen application disrupts the carbon–nitrogen metabolic balance, inhibits the transport of assimilates to storage organs, and leads to excessive accumulation of nitrate nitrogen [16], thereby affecting the quality and safety of agricultural products. Furthermore, a nitrogen surplus can produce the potent greenhouse gas N_2_O through nitrification–denitrification processes [17]. It also disrupts the structure of the soil’s microbial community, reducing the abundance of nitrogen-fixing bacteria and phosphorus-solubilizing bacteria while increasing the proportion of pathogenic bacteria. Thereby, this inhibits soil enzyme activity, disrupts aggregate stability, and ultimately leads to a decline in soil quality and low and unstable crop productivity [18,19]. In addition, during the nitrogen absorption and transport process, an excessive nitrogen supply can interfere with the active absorption of nitrogen by the root system [20], thereby affecting nitrogen use efficiency and productivity. Therefore, optimizing nitrogen application rates is of great practical significance for improving nitrogen use efficiency and reducing environmental risks.

Although sub-surface drip irrigation technology has been proven to significantly increase crop yields and water use efficiency [21], current vegetable production practices still excessively pursue a high yield, neglecting the negative impact of unreasonable water and nitrogen management on quality. At the same time, excessive water and nitrogen supplies increase production costs and reduce economic efficiency. Therefore, based on a four-year consecutive field experiment, this study systematically investigates the effects of water–nitrogen coupling on onion quality, yield, water–nitrogen use efficiency, and economic benefits. A water–nitrogen economic benefit model was constructed to provide a scientific basis for water-saving and fertilizer-saving cultivation of specialty vegetables in the Hexi Oasis agricultural area. This research holds significant practical value for promoting the sustainable development of oasis agriculture along the Belt and Road initiative.

## 2. Results and Analysis

### 2.1. The Effect of Water–Nitrogen Interaction on Onion Quality

Reasonable water and nitrogen management is the foundation for high-yield, high-quality crops. The results of the analysis of variance showed that the single factors of irrigation amount and nitrogen application rate, as well as their interaction, had a significant effect on onion quality (*p* < 0.05) (Table 1). The four-year data all show that under the same nitrogen application rate, bulb fresh weight, diameter, length, soluble sugar content, and vitamin C content all increased with an increasing irrigation amount. When averaged across the same irrigation levels, the average increases from W1 to W2 were 85.65%, 19.15%, 20.88%, 19.34%, and 14.44%, respectively. From W2 to W3, the average increases were 15.43%, 8.78%, 8.92%, 3.26%, and 1.66%, respectively. It can be seen that as the irrigation amount increases, the increase in soluble sugar and vitamin C content in the bulbs gradually diminished. Additionally, at the W3 irrigation level, bulb onion oil, soluble protein, and propionic acid showed a decreasing trend compared to W2, with decreases of 3.24%, 8.47%, and 6.09%, respectively, indicating an inhibitory effect. The nitrogen application rate also significantly affected the onion quality. When averaged across the same nitrogen application level, from N0 to N1, the bulb fresh weight, diameter, length, onion oil, soluble sugar, soluble protein, vitamin C, and propionic acid increased by 48.02%, 17.61%, 15.93%, 8.96%, 5.76%, 13.66%, 7.50%, and 10.60%, respectively. From N1 to N2, these parameters further increased by 18.48%, 8.82%, 9.28%, 15.59%, 5.30%, 11.25%, 10.16%, and 18.47%, respectively. In contrast, from N2 to N3, these parameters decreased by 6.05%, 4.22%, 4.84%, 6.16%, 15.39%, 3.08%, 3.77%, and 5.08%, respectively. It can be seen that although an increasing nitrogen application rate can improve the appearance and nutritional quality of onion, there is a threshold for the promoting effect, and beyond this threshold, the bulb quality exhibits a declining trend. Similarly, the water–nitrogen coupling effect also significantly influenced the onion quality. Results from the four-year experiment indicated that medium-to-high level combinations of water and nitrogen were most beneficial for onion quality accumulation. Due to differences in climate and irrigation timing across the experimental years, the N3W3 treatment in 2020 yielded the highest bulb fresh weight, diameter, and length, at 412.5 g, 93.25 mm, and 109.18 mm, respectively. Meanwhile, the N2W3 treatment had the highest contents of onion oil, soluble sugars, soluble proteins, vitamin C, and propionic acid, at 0.42%, 180.45 mg·g^−1^, 22.98 mg·g^−1^, 20.70 mg·100g^−1^, and 0.30 mg·g^−1^, respectively. From 2021 to 2023, the N2W3 treatment yielded the best bulb fresh weight, diameter, length performance, and highest soluble sugar content, averaging 393.67 g, 92.12 mm, 97.72 mm, and 181.11 mg·g^−1^, respectively. Meanwhile, the N2W2 treatment yielded the highest levels of onion oil, soluble protein, vitamin C, and propionic acid content, averaging 0.40%, 22.33 mg·g^−1^, 20.59 mg·100g^−1^, and 0.30 mg·g^−1^, respectively.

### 2.2. The Effect of Water–Nitrogen Interaction on Onion Yield

The effects of water–nitrogen interactions on the onion yield are shown in Table 2. As shown in Table 2, irrigation, nitrogen application, and water–nitrogen interaction significantly affected the onion yield (*p* < 0.05). Variance analysis revealed that in the Hexi Oasis agricultural region, where annual rainfall is low and evaporation is high, irrigation is the primary factor affecting onion yield, followed by nitrogen application and water–nitrogen interaction. When averaged across the same irrigation levels, onion yields increased by 65.06%, 72.51%, 87.05%, and 118.77% from W1 to W2 in 2020 to 2023, with an average increase of 85.85% across the four growing seasons. From W2 to W3, yields increased by 9.86%, 10.11%, 18.71%, and 20.50%, respectively, with an average increase of 14.79% across the four growing seasons. This indicates that an increasing irrigation amount can significantly enhance the onion yield, but the yield-increasing effect gradually diminishes as the irrigation amount increases. The nitrogen application rate is a secondary factor affecting the onion yield. When averaged across the same nitrogen application level, from 2020 to 2023, the onion yield increased by 35.28%, 43.93%, 55.88%, and 62.81% from N0 to N1, with an average increase of 49.47% across the four growing seasons. From N1 to N2, yields increased by 10.39%, 22.16%, 24.47%, and 16.61%, respectively, with an average increase of 18.41% across the four growing seasons. However, from N2 to N3, yields decreased by 2.44%, 7.41%, 9.61%, and 4.67%, respectively, with an average decrease of 6.03% across the four growing seasons. It is evident that the application of nitrogen fertilizer has a threshold effect on the onion yield. Exceeding the nitrogen application level of N2 actually inhibits onion growth and reduces the yield. The water–nitrogen interaction also significantly affected the onion yield. In 2020, the N3W3 treatment yielded the highest output at 136.93 t·ha^−1^. In 2021, 2022, and 2023, the N2W3 treatment yielded the highest output, reaching 134.41, 125.94, and 132.26 t·ha^−1^, respectively. Across all four growing seasons, the N0W1 treatment exhibited the lowest yield, with values of 55.25, 46.77, 36.98, and 28.26 t·ha^−1^ from 2020 to 2023, respectively, averaging only 41.81 t·ha^−1^.

### 2.3. The Effect of Water–Nitrogen Interaction on Onion Water–Nitrogen Use Efficiency

Appropriate water and nitrogen management is the foundation for efficient water and nitrogen use in crops. Irrigation amount, nitrogen application rate, and water–nitrogen interaction effects have a significant impact on onion water–nitrogen use efficiency, and the level of impact varies across different years (Table 3). When averaged across the same nitrogen application rate, onion water consumption increased with the increasing nitrogen application rate, and the WUE and IWUE exhibited a parabolic trend with the increasing nitrogen application rate. From N0 to N1, onion ET, WUE, and IWUE increased by an average of 11.79%, 33.70%, and 49.56%, respectively, over four years. From N1 to N2, the corresponding increases were 2.17%, 14.88%, and 17.29%, respectively. From N2 to N3, only the ET increased by 1.84%, while the WUE and IWUE decreased by 7.82% and 6.48%, respectively. It can be seen that excessive nitrogen application limits the efficient use of water by onions, leading to a decrease in the WUE and IWUE. When averaged across the same irrigation level, the onion ET increases with an increasing amount of irrigation. From W1 to W2, the ET increased by an average of 15.36% over four years, and from W2 to W3, it increased by an average of 14.84%. Due to interannual variations in precipitation and other farm climate conditions, changes in the WUE and IWUE varied. Specifically, in 2020 and 2021, both the WUE and IWUE at W3 decreased by 4.52% and 6.62%, respectively, compared to W2, while the irrigation WUE decreased by 6.21% and 6.41%, respectively. In contrast, in 2022 and 2023, both the WUE and IWUE improved with increasing irrigation. From W1 to W2, the WUE increased by 64.12% and 54.04%, while the IWUE increased by 84.42% and 80.17%. From W2 to W3, the increases were 2.69% and 0.90% for the WUE, and 9.07% and 2.42% for the IWUE. It can be seen that as the irrigation amount increases, the improvement in the WUE becomes significantly diminished, and excessive irrigation also limits the efficient utilization of water by onions. Water and nitrogen interact with and promote each other. The N3W3 treatment consumed the highest ET, with a four-year average of 785.72 mm. The N2W2 treatment had the highest WUE and IWUE, with four-year averages of 17.66 kg·m^−3^ and 23.15 kg·m^−3^, respectively. PFP_N_ and AUE_N_ varied across different experimental years. In 2020, 2022, and 2023, the N1W3 treatment achieved the highest PFP_N_, with an average of 557.86 kg·kg^−1^ over the three years. The N2W3 treatment exhibited the highest AUE_N_, averaging 225.40 kg·kg^−1^. In 2021, the N2W3 treatment recorded the highest PFP_N_ and AUE_N_, reaching 509.13 and 241.15 kg·kg^−1^, respectively. Meanwhile, data analysis revealed that under the same irrigation amount, PFP_N_ showed a significantly negative correlation with the nitrogen application rate, while AUE_N_ is positively correlated with the nitrogen application rate. When nitrogen application rates were the same, PFP_N_ was positively correlated with the irrigation amount. Specifically, from W1 to W2, PFP_N_ increased by an average of 90.70% over four years, and from W2 to W3, it increased by an average of 12.56%.

### 2.4. The Effect of Water–Nitrogen Interaction on Onion Economic Benefits

The input costs and economic benefits of onion under different water and nitrogen treatments from 2020 to 2023 are shown in Table 4. An increase in irrigation amount and nitrogen application rate resulted in a rising trend in total input costs. Data from the four-year experiment indicated that the N3W3 treatment had the highest expenses for water input, fertilizer input, and total investment, averaging 197.76, 1026.12, and 6476.90 USD·ha^−1^, respectively, representing increases of 17.65% to 42.85%, 5.05% to 31.67%, and 0.46% to 4.96% compared to other treatments. Irrigation amount, nitrogen application rate, and the interaction effect of water and nitrogen significantly influenced the economic benefits and net profit. The economic benefits and net profit exhibit a positive correlation with the irrigation amount. When averaged across the same irrigation level, the economic benefits and net profit increased by an average of 83.86% and 210.82%, respectively, from W1 to W2 over four years, and by 14.84% and 21.91%, respectively, from W2 to W3. This indicates that although the increasing irrigation amount raises farm input costs, it also significantly increases the onion yield, thereby improving the economic benefits and net profit. Under the same nitrogen application rate, the economic benefits and net profit increased by an average of 49.11% and 102.15%, respectively, from N0 to N1 over four years, and by 18.92% and 29.35%, respectively, from N1 to N2. However, from N2 to N3, they decreased by 6.39% and 9.60%, respectively. This indicates that excessive nitrogen application leads to a decline in onion economic benefits and net profit. This is because nitrogen fertilizer has a limited effect on the increasing onion yield, and high nitrogen levels result in increased farm inputs, thereby reducing the net profit. During the four-year experiment, the highest economic benefits and net profit of onion did not occur under the same treatment. In 2020, the N3W3 treatment yielded the highest economic benefit and net profit, reaching 20,376.25 and 14,320.79 USD·ha^−1^, respectively. From 2021 to 2023, the N2W3 treatment yielded the highest economic benefit and net profit, averaging 28,449.47 and 21,881.53 USD·ha^−1^, respectively. In contrast, the N0W1 treatment maintained the lowest economic benefit and net profit over the four years, averaging only 8199.15 and 2028.39 USD·ha^−1^, respectively.

### 2.5. Water and Nitrogen Coupling Model and Scheme Optimization for Drip Irrigation Under Mulch

#### 2.5.1. Water and Nitrogen Coupling Model Equation

Based on the four-year economic benefit data from the water–nitrogen coupling experiment, we conducted a bivariate quadratic regression simulation and obtained the regression model for onion economic benefits (*y*), irrigation amount coded values (*x*_1_), and nitrogen application rate coded values (*x*_2_) for 2020–2023:(1)$$y_{2020}=18363.15+4516.80x_{1}+547.45x_{2}-2826.37x_{1}^{2}-976.51x_{2}^{2}+1105.09x_{1}x_{2}$$(2)$$y_{2021}=26250.53+6690.98x_{1}+1260.36x_{2}-4150.46x_{1}^{2}-3000.21x_{2}^{2}+2048.93x_{1}x_{2}$$(3)$$y_{2022}=26098.45+7758.62x_{1}+1168.52x_{2}-4272.54x_{1}^{2}-3401.89x_{2}^{2}+1636.33x_{1}x_{2}$$(4)$$y_{2023}=22190.89+7621.38x_{1}+934.12x_{2}-4011.79x_{1}^{2}-1846.05x_{2}^{2}+774.23x_{1}x_{2}$$(5)$$y_{2020-2022}=23570.70+6322.13x_{1}+1001.11x_{2}-3749.79x_{1}^{2}-2459.53x_{2}^{2}+1596.78x_{1}x_{2}$$

Significance tests were conducted on Equations (1)–(5), yielding the coefficient of the determination values of 0.984, 0.987, 0.985, 0.994, and 0.988, respectively. These results indicate that the regression model has high predictive accuracy and can effectively simulate the economic benefits of onion. After testing, F2020 = 36.94, *p* < 0.01; F2021 = 45.79, *p* < 0.01; F2022 = 39.27, *p* < 0.01; F2023 = 108.35, *p* < 0.01; and F2020–2022 = 48.59, *p* < 0.01, indicating that the regression relationship reaches a highly significant level, and the regression model fits well, effectively reflecting the relationship between economic benefits and irrigation amount and nitrogen application rate.

#### 2.5.2. Model Validation

After the calibration of the model parameters using experimental data from 2020 to 2022, the predictive accuracy of the regression model was validated by using the 2023 dataset as an independent validation set. The results showed a strong agreement between the predicted and observed economic benefits (MRE = 9.53%, R^2^ = 0.988, RMSE = 1755.9 USD·ha^−1^), confirming the robustness of the model within the experimental conditions. However, due to differences in soil properties, climatic conditions, and management practices, the transferability of this model in other regions or under significant climate change scenarios may be limited. Future research should incorporate multi-site and multi-year data to enhance the generalizability of the model.

#### 2.5.3. Single-Factor Effect Analysis

To further analyze the impact of individual factors—irrigation amount and nitrogen application rate—on economic benefits following optimization, a dimensionality reduction method was adopted to set any one of the two factors in the regression equations (1) to (4) to a zero level. The single-factor economic benefit effect functions for the irrigation amount (*y_w_*) and nitrogen application rate (*y_n_*) were derived as follows:(6)$$2020\;\mathrm{year}\;\;\;\;y_{w}=18363.15+4516.80x_{1}-2826.37x_{1}^{2}$$(7)$$y_{n}=18363.15+547.45x_{2}-976.51x_{2}^{2}$$(8)$$2021\;\mathrm{year}\;\;\;\;y_{w}=26250.53+6690.98x_{1}-4150.46x_{1}^{2}$$(9)$$y_{n}=26250.53+1260.36x_{2}-3000.21x_{2}^{2}$$(10)$$2022\;\mathrm{year}\;\;\;\;y_{w}=26098.45+7758.62x_{1}-4272.54x_{1}^{2}$$(11)$$y_{n}=26098.45+1168.52x_{2}-3401.89x_{2}^{2}$$(12)$$2023\;\mathrm{year}\;\;\;\;y_{w}=22190.89+7621.38x_{1}-4011.79x_{1}^{2}$$(13)$$y_{n}=22190.89+934.12x_{2}-1846.05x_{2}^{2}$$

The impact of each factor on the economic benefits is illustrated in Figure 1. It can be observed that the economic benefits of both water and nitrogen as single factors exhibit a parabolic pattern. Within the range of experimental design levels, the influence of water and nitrogen as single factors on the economic benefits is positive. However, as the irrigation amount and nitrogen application rate increase, the economic benefits gradually diminish, which conforms to the diminishing returns effect; there is a maximum point of economic benefit. Due to variations in microclimate conditions such as interannual precipitation, the maximum value differs across experimental years. In 2020, when *x*_1_ = 0.7990, the corresponding irrigation amount was 565.7 mm, yielding the maximum onion economic benefit of 20,167.7 USD·ha^−1^. When *x*_2_ = 0.2803, the corresponding nitrogen application rate was 282.5 kg·ha^−1^, yielding the maximum onion economic benefit of 18,439.9 USD·ha^−1^. In 2021, when *x*_1_= 0.8061, the corresponding irrigation amount was 621.2 mm, yielding the maximum onion economic benefit of 28,947.2 USD·ha^−1^. When *x*_2_ = 0.2100, the corresponding nitrogen application rate was 277.9 kg·ha^−1^, yielding the maximum onion economic benefit of 26,382.9 USD·ha^−1^. In 2022, when *x*_1_ = 0.9080, the corresponding irrigation amount was 563.8 mm, yielding the maximum onion economic benefit of 29,620.7 USD·ha^−1^. When *x*_2_ = 0.1717, the corresponding nitrogen application rate was 275.3 kg·ha^−1^, yielding the maximum onion economic benefit of 26,198.8 USD·ha^−1^. In 2023, when *x*_1_ = 0.9499, the corresponding irrigation amount was 655.4 mm, yielding the maximum onion economic benefit of 25,810.6 USD·ha^−1^. When *x*_2_ = 0.2530, the corresponding nitrogen application rate was 280.7 kg·ha^−1^, yielding the maximum onion economic benefit of 22,309.1 USD·ha^−1^. By analyzing the trends of water and nitrogen factor variation curves from 2020 to 2023, it is demonstrated that when the irrigation amount or nitrogen application rate falls below the maximum point, a positive effect is observed. The economic benefits of onion increase with the rising irrigation amount and nitrogen application rate. Conversely, when the irrigation amount or nitrogen application rate exceeds the maximum point, a negative effect is exhibited. That is, the economic benefits of onion decline as the irrigation amount and nitrogen application rate continue to increase.

#### 2.5.4. Single-Factor Marginal Effect Analysis

Marginal economic efficiency reflects the impact of optimal input levels and unit-level input variations on the rate of increase or decrease in economic benefits. The marginal economic efficiency of each factor at different levels can be derived by taking the first-order partial derivatives of regression sub-models (6) to (13). The marginal effect equations for the nitrogen application rate and irrigation amount are as follows:(14)$$2020\;\mathrm{year}\;\;\;dy_{w}/dx_{1}=4516.80-5652.73x_{1}$$(15)$$dy_{n}/dx_{2}=547.45-1953.01x_{2}$$(16)$$2021\;\mathrm{year}\;\;\;dy_{w}/dx_{1}=6690.98-8300.91x_{1}$$(17)$$dy_{n}/dx_{2}=1260.36-6000.42x_{2}$$(18)$$2022\;\mathrm{year}\;\;\;dy_{w}/dx_{1}=7758.62-8545.09x_{1}$$(19)$$dy_{n}/dx_{2}=1168.5-6803.79x_{2}$$(20)$$2023\;\mathrm{year}\;\;\;dy_{w}/dx_{1}=7621.38-8023.59x_{1}$$(21)$$dy_{n}/dx_{2}=934.12-3692.11x_{2}$$

By plotting the corresponding marginal effect diagram based on the single-factor marginal functions (Figure 2), it can be observed that the marginal economic benefit of onion diminishes as the irrigation amount and nitrogen application rate increase. A positive value on the vertical axis indicates that the factor improves the water and nitrogen environment for onion growth, thereby increasing the profits. Conversely, a vertical axis of less than zero indicates that it will lead to a reduction in the economic benefits of onion. The four-year field experiments consistently demonstrated a pattern: when it was less than the zero value point, the economic benefits increased with the increase in the irrigation amount and nitrogen application rate, indicating that irrigation and nitrogen application are conducive to increasing the onion yield and economic benefits. Conversely, when it was greater than the zero value point, increased irrigation and nitrogen application exhibit an inhibitory effect. Specifically, the yield-enhancing impact of additional irrigation and nitrogen diminishes markedly, while the corresponding farmland input costs rise. This leads to diminished economic returns, and in some instances, farmland input costs may even exceed the yield income. In 2020, when *x*_1_ ≤ 0.7990, the economic benefits increased with the increase in irrigation amount, when *x*_1_ > 0.7990, the economic benefits decreased with the increase in irrigation amount, when *x*_2_ ≤ 0.2803, the nitrogen application enhanced onion’s economic benefits, and when *x*_2_ > 0.2803, the nitrogen application inhibited improvement in onion’s economic benefits. In 2021, when *x*_1_ ≤ 0.8061, the economic benefits increased with the increase in the irrigation amount, when *x*_1_ > 0.8061, the economic benefits decreased with the increase in the irrigation amount, when *x*_2_ ≤ 0.2100, the nitrogen application enhanced onion’s economic benefits, and when *x*_2_ > 0.2100, the nitrogen application inhibited the improvement in onion economic benefits. In 2022, when *x*_1_ ≤ 0.9080, the economic benefits increased with the increase in irrigation amount, when *x*_1_ > 0.9080, the economic benefits decreased with the increase in irrigation amount, when *x*_2_ ≤ 0.1717, the nitrogen application enhanced onions’ economic benefits, and when *x*_2_ > 0.1717, the nitrogen application inhibited the improvement of onions’ economic benefits. In 2023, when *x*_1_ ≤ 0.9499, the economic benefits increased with the increase in irrigation amount, when *x*_1_ > 0.9499, the economic benefits decreased with the increase in irrigation amount, when *x*_2_ ≤ 0.2530, the nitrogen application enhanced onions’ economic benefits, and when *x*_2_ > 0.2530, the nitrogen application inhibited the improvement in onions’ economic benefits.

#### 2.5.5. Analysis of the Interaction Effect of Water and Nitrogen Factors

The economic benefits of onion exhibit a synergistic effect between water and nitrogen within a certain range. Figure 3 presents a three-dimensional interaction diagram illustrating the combined effects of the irrigation amount and nitrogen application rate on onion’s economic benefits. As shown in the figure, it reveals that both the irrigation amount and nitrogen application rate exhibit parabolic relationships with onion’s economic benefits. At a constant irrigation amount, the economic benefit initially increases and then decreases with the rising nitrogen application rate. Similarly, at a constant nitrogen application rate, the economic benefit initially increases and then decreases with a rising irrigation amount. In 2020, the maximum simulated onion economic benefit was 20,756.7 USD·ha^−1^, corresponding to *x*_1_ = 0.9601 and *x*_2_ = 0.8236, i.e., an irrigation amount of 579.8 mm and nitrogen application rate of 318.4 kg·ha^−1^. In 2021, the maximum simulated onion economic benefit was 29,718.8 USD·ha^−1^, corresponding to *x*_1_ = 0.9369 and *x*_2_ = 0.5299, equivalent to an irrigation amount of 633.8 mm and a nitrogen application rate of 2989.0 kg·ha^−1^. In 2022, the maximum simulated onion economic benefit was 30,163.4 USD·ha^−1^, corresponding to *x*_1_ = 0.9863 and *x*_2_ = 0.4090, i.e., an irrigation amount of 570.6 mm and a nitrogen application rate of 291.0 kg·ha^−1^. In 2023, the maximum simulated onion economic benefit was 26,195.8 USD·ha^−1^, corresponding to *x*_1_ = 0.9944 and *x*_2_ = 0.4616, which was equivalent to an irrigation amount of 659.8 mm and a nitrogen application rate of 294.5 kg·ha^−1^. The results indicate a strong coupling effect between the irrigation amount and the nitrogen application rate. The optimal allocation of these two factors is crucial for enhancing onion’s economic benefits, with the highest yields being consistently observed at medium-to-high levels of both irrigation and nitrogen application.

#### 2.5.6. Optimization of Combination Schemes

To derive the optimal water–nitrogen combination schemes for achieving different target economic benefits of onion under optimized water–nitrogen management strategies, the frequency method was employed to further analyze Equations (1) to (4). Six equidistant levels between 1.0 and −1.0 (1.0, 0.8, 0.6, 0.4, 0.2, 0.0, −0.2, −0.4, −0.6, −0.8, −1.0) were selected. Through a simulation-based solution, 121 combination schemes were obtained. The optimized combination scheme is presented in Table 5, with the 95% confidence interval derived via SPSS analysis. As shown in Table 5, under optimized water and nitrogen management strategies, the following water–nitrogen combination schemes yielding different target economic benefits over four years were obtained: (1) For a target onion economic benefit ranging from 17,000 to 19,000 USD·ha^−1^, in 2020, the recommended irrigation amounts were 416.5–465.3 mm and the nitrogen application rates were 146.0–204.5 kg·ha^−1^. In 2021, the irrigation amounts ranged from 295.5 to 323.9 mm, with the nitrogen application rates being between 127.1 and 253.1 kg·ha^−1^. In 2022, the irrigation amounts ranged from 270.5 to 314.3 mm, with the nitrogen application rates being between 91.5 and 259.3 kg·ha^−1^. In 2023, the irrigation amounts ranged from 379.4 to 401.0 mm, with the nitrogen application rates being between 139.2 and 256.8 kg·ha^−1^. (2) For a target onion economic benefit ranging from 19,000 to 21,000 USD·ha^−1^, in 2020, the recommended irrigation amounts were 503.1–534.9 mm and the nitrogen application rates were 219.8–259.0 kg·ha^−1^. In 2021, the optimal irrigation amounts were 332.4–367.9 mm and the nitrogen application rates were 117.1–235.7 kg·ha^−1^. In 2022, the irrigation amounts were 300.7–335.0 mm and the nitrogen application rates were 127.2–253.1 kg·ha^−1^. In 2023, the irrigation amounts were 422.3–453.5 mm and the nitrogen application rates were 140.2–206.6 kg·ha^−1^. (3) For a target onion economic benefit ranging from 21,000 to 23,000 USD·ha^−1^, in 2021, the optimal irrigation amounts were 365.6–459.3 mm and the nitrogen application rates were 120.9–234.5 kg·ha^−1^. In 2022, the irrigation amounts were 340.2–381.1 mm and the nitrogen application rates were 124.3–235.1 kg·ha^−1^. In 2023, the irrigation amounts were 589.4–620.8 mm and the nitrogen application rates were 127.4–236.1 kg·ha^−1^. (4) For a target onion economic benefit ranging from 23,000 to 25,000 USD·ha^−1^, in 2021, the irrigation amounts were 441.6–518.9 mm and the nitrogen application rates were 110.6–193.7 kg·ha^−1^. In 2022, the irrigation amounts were 387.1–463.7 mm and the nitrogen application rates were 112.6–216.5 kg·ha^−1^. In 2023, the irrigation amounts were 477.3–540.4 mm and the nitrogen application rates were 123.4–222.9 kg·ha^−1^. (5) For a target onion economic benefit ranging from 25,000 to 27,000 USD·ha^−1^, in 2021, the irrigation amounts were 470.4–536.2 mm and the nitrogen application rates were 152.6–225.8 kg·ha^−1^. In 2022, the irrigation amounts were 430.5–491.9 mm and the nitrogen application rates were 138.4–222.4 kg·ha^−1^. In 2023, the irrigation amounts were 542.0–583.7 mm and the nitrogen application rates were 161.9–224.0 kg·ha^−1^. (6) For a target onion economic benefit ranging from 27,000 to 29,000 USD·ha^−1^, in 2021, the irrigation amounts were 529.6–572.1 mm and the nitrogen application rates were 203.1–257.7 kg·ha^−1^. In 2022, the irrigation amounts were 469.9–511.4 mm and the nitrogen application rates were 194.8–254.0 kg·ha^−1^. In 2023, the irrigation amounts were 609.4–640.3 mm and the nitrogen application rates were 226.4–275.2 kg·ha^−1^.

## 3. Discussion

### 3.1. Effects of Water–Nitrogen Interactions on Onion Quality

Water and nitrogen are key limiting factors in enhancing crop productivity, with both participating in the physiological metabolic processes of crops. Water stress disrupts normal metabolic activities within crop cells, upsetting their metabolic homeostasis and inducing the activation of osmotic regulation mechanisms [22,23]. Furthermore, water conditions influence the allocation of photosynthetic products within leaves, thereby regulating the formation of crop quality [24]. Nitrogen, as a fundamental constituent element of biomolecules such as nucleic acids, amino acids, and proteins [25], directly influences crop growth and development through its supply levels. However, excessive nitrogen application disrupts ionic homeostasis within plants, inhibits root growth, impairs photosynthetic systems, and may even induce NH_4_^+^ toxicity [26,27,28]. Concurrently, it can precipitate environmental issues, including acid rain and nitrogen leaching. As a shallow-rooted crop, onion is particularly sensitive to water and nitrogen. Research by Wakchaure et al. [29,30] indicates that water stress activates the metabolic regulatory mechanisms in onions, significantly increasing the content of total soluble sugars and proteins. Conversely, under waterlogged conditions, their antioxidant systems become compromised, carbohydrate metabolism is disrupted, and protein and soluble solids content markedly decrease, while pyruvate content increases significantly. This study found that irrigation levels significantly influenced onion quality. Although irrigation at 100% ET_c_ promoted bulbification and increased the soluble sugar content compared to 85% ET_c_, it resulted in reductions in the onion oil, soluble protein, vitamin C, and pyruvate contents. This aligns with the findings of Wakchaure et al. [29,30], which indicate that excessive irrigation inhibits onion quality formation. The nitrogen application rate also significantly regulates onion quality. The results of this experiment indicate that when nitrogen application exceeded 264 kg·ha^−1^, all onion quality indicators showed a declining trend, though the extent of reduction varied. This finding aligns with the conclusion proposed by Han et al. [31] that moderate nitrogen reduction can enhance onion quality. However, due to differences in field management, ecological climate zones, and varieties, the effects of nitrogen may exhibit regional variation. It is noteworthy that synergistic and antagonistic effects exist between water and nitrogen, making the optimization of water–nitrogen ratios a key measure for improving onion quality. In this experiment, irrigation at 100% ET_c_ combined with nitrogen application rates of 264–330 kg·ha^−1^ proved most conducive to root nutrient uptake and metabolic homeostasis, thereby achieving optimal quality. This aligns with findings from Rani et al. [32], who also reported that water–nitrogen synergy optimized onion quality. Consequently, rational water–nitrogen management forms the foundation for sustaining normal physiological activity in onions, significantly enhancing bulb quality by promoting root nutrient uptake and metabolic processes.

### 3.2. Effects of Water–Nitrogen Interactions on Onion Yield and Water–Nitrogen Use Efficiency

High-efficiency crop production constitutes a pivotal component of sustainable agricultural development. As a typical water-intensive vegetable, onion yield is intrinsically linked to water and nitrogen management practices. Research indicates that inadequate water and nitrogen inputs significantly constrain onion yield and water–nitrogen use efficiency, thereby jeopardizing the security of ‘vegetable basket’ supplies. Conversely, excessive inputs lead to a decline in water–nitrogen use efficiency, with over-application of nitrogen fertilizers further exacerbating issues such as soil residue accumulation and environmental pollution [33]. Therefore, optimizing the water-to-nitrogen ratio is the key approach to achieving the synergistic effect of ‘regulating fertilizer with water and promoting water use with fertilizer’, thereby balancing both the yield and the economic benefits. Crop water requirements can be determined by referencing evapotranspiration and effective rainfall. When the irrigation amount remains within the optimal range, the yield increases with higher water application; however, beyond a certain threshold, the yield-enhancing effect diminishes or even reverses into yield reduction [34]. The application of nitrogen fertilizer also exhibits a threshold effect, where excessive nitrogen application inhibits crop growth [35]. In the Hexi Corridor region, current nitrogen application rates in farmland are generally excessive, resulting in substantial residual soil nitrogen [36]. Previous studies have confirmed that water and nitrogen management significantly impact the onion yield. Piri et al. [3] found that the irrigation amount, nitrogen application rate, and their interaction significantly influenced the onion bulb morphology and yield, with the yield-increasing effect diminishing as input levels continued to rise. Mubarak et al. [37] indicated that water is the primary limiting factor for the onion yield, exerting a greater influence than nitrogen fertilizer application. Tagay et al. [38] further confirmed that the irrigation frequency and nitrogen application rate exert significant regulatory effects on the yield. The findings of this study align with previous research, indicating that under the same nitrogen application levels, the yield exhibits a linear positive correlation with the irrigation amount. This phenomenon may arise because the increasing irrigation promotes the dissolution of the soil’s nitrogen and its transport to the root zone, thereby enabling efficient nitrogen use in photosynthesis and protein synthesis. This process helps prevent the carbon–nitrogen metabolic imbalance that is typically induced by drought conditions. Conversely, under the same irrigation levels, the yield exhibits a parabolic relationship with the nitrogen application rate. When the nitrogen application exceeds 264 kg·ha^−1^, the yield-enhancing effect of the nitrogen gradually diminishes and may even become inhibitory. This phenomenon may be attributed to excessive nitrogen application inducing lipid peroxidation in root cell membranes, thereby reducing the water and nutrient uptake capacity. Concurrently, the nitrification process of excess ammonium nitrogen releases H^+^ ions, which inhibit soil microbial activity and reduce the availability of the trace elements. The significance of the water–nitrogen interaction effect varied across years, which may be attributed to interannual climatic variability, particularly differences in rainfall distribution and temperature. In years with more uniform rainfall, the synergistic effect of water and nitrogen may be less pronounced, whereas under more variable conditions, their interaction becomes more critical for yield stabilization.

Water–nitrogen use efficiency serves as a pivotal indicator for evaluating high-yield, efficient crop production systems. Optimizing the water and nitrogen synergistic management is of significant importance for enhancing agricultural quality and productivity. Research has confirmed that water–nitrogen interactions significantly influence water–nitrogen use efficiency. Piri et al. [3] found that under the same irrigation levels, increasing the nitrogen application enhances the onion WUE. Conversely, under the same nitrogen application levels, the IWUE exhibits a parabolic relationship with the irrigation amount. Research by Deng et al. [39] indicated that moderate water stress not only maintains a stable yield but also simultaneously enhances both the WUE and the IWUE. The results of this study indicate that different water and nitrogen management strategies significantly influence the water consumption characteristics of onion. Specifically, the sufficient-water and high-nitrogen treatment exhibited the highest water consumption, averaging 785.72 mm over four years. Both the WUE and the IWUE exhibited a unimodal curve with water and nitrogen inputs, peaking at 85% ET_c_ irrigation levels with 264 kg·ha^−1^ nitrogen application, averaging 17.66 kg·m^−3^ and 23.15 kg·m^−3^ over four years, respectively. This phenomenon indicates that water–nitrogen management possesses an optimal threshold. When inputs fall below this threshold, water–nitrogen interactions can synergistically enhance the WUE. However, exceeding the threshold leads to a decline in WUE with additional input. This finding aligns with previous reports [39]. Concurrently, research indicates that PFP_N_ exhibits a negative correlation with the nitrogen application rate whilst showing a positive correlation with the irrigation amount [40]. This study confirms this pattern: namely, under the same nitrogen application levels, PFP_N_ increases with a higher irrigation amount, whereas under the same irrigation levels, it decreases with a higher nitrogen application rate. Among all treatments, the sufficient-water and low-nitrogen treatment yielded the highest PFP_N_, ranging from 501.45 to 599.92 kg·kg^−1^. This study also found that the AUE_N_ exhibits a similar pattern to the water–nitrogen response and PFP_N_. Specifically, under the same nitrogen application levels, increasing irrigation enhances the AUE_N_, whereas under the same irrigation levels, increasing nitrogen application diminishes the AUE_N_. This conclusion was corroborated in studies by Banik [41] and Rani [42]. Therefore, optimizing the water-to-nitrogen ratio can effectively coordinate the water and nitrogen requirements of onion, promoting efficient nitrogen uptake and utilization. This approach simultaneously enhances both the PFP_N_ and AUE_N_ in onion.

### 3.3. Effects of Water–Nitrogen Interactions on Onion’s Economic Benefits

The economic benefits of agricultural production result from the combined effects of yield, input costs, and market returns. As the nitrogen application rate and irrigation amount increase, the farmland input costs rise linearly. However, the yield-enhancing effect of sustained water and nitrogen inputs gradually diminishes. Indeed, excessive water and nitrogen inputs lead to farmland input costs surpassing the crop’s economic benefits, resulting in negative profitability. This study found that increasing the irrigation amount and nitrogen application rate raised the total input costs for farmland. Over the four years, the sufficient-water and high-nitrogen treatment consistently yielded the highest costs, averaging 6476.90 USD·ha^−1^. However, the yield returns from this treatment were lower than those from the sufficient-water and medium-nitrogen treatment, resulting in lower economic benefits and net profits than the treatment with sufficient-water and medium-nitrogen. As the experiment progressed, onion yields under the zero-nitrogen treatment continued to decline, while the water input and other input costs exceeded yield returns, resulting in negative net profits in the low-water and zero-nitrogen treatment by 2023. These findings align with the conclusions of studies by Zhang [43] and Piri [44], which confirmed that increasing irrigation and nitrogen application can enhance crop yields and thereby improve the economic benefits. However, a threshold exists between the yield and water–nitrogen inputs. Beyond this threshold, the yield increases diminish markedly, and crop growth may even be inhibited, leading to reduced yields and lower farmland net profits. Therefore, optimizing the irrigation amount and nitrogen application rate enhances the crop nitrogen uptake, creates a water–nitrogen synergistic effect, and reduces the farmland input costs, while achieving increased yields and income. Li et al. [45] established a water–nitrogen coupling model for sunflowers, revealing that the irrigation amount, nitrogen application rate, and economic benefits conform to a binary quadratic regression model. The economic benefits exhibit a parabolic trend with increasing water and nitrogen inputs. Furthermore, this model accurately predicted the irrigation amount and nitrogen application rate corresponding to different target economic benefits. Pan et al. [21] employed a water–nitrogen coupling model for multiple regression analysis, revealing that a sufficient-water and medium-nitrogen strategy can simultaneously enhance the seed maize yield and water–nitrogen productivity. The binary quadratic regression equation model of water–nitrogen economic benefits established in this study exhibits high predictive accuracy, with regression relationships reaching extremely significant levels. It can more effectively simulate onion’s economic benefits and elucidate the relationship between the economic benefits and the irrigation amount and nitrogen application rate. Due to interannual climatic variations, the water and nitrogen application rates showed slight differences. In 2021, irrigation amounts ranged from 529.6 to 527.1 mm, with nitrogen application rates being between 203.1 and 257.7 kg·ha^−1^. In 2022, irrigation amounts ranged from 469.9 to 511.4 mm, with nitrogen application rates being from 194.8 to 254.0 kg·ha^−1^. In 2023, irrigation amounts ranged from 609.4 to 640.3 mm, with nitrogen application rates being from 226.4 to 275.2 kg·ha^−1^. Across all three years, onion target economic benefits reached 27,000–29,000 USD·ha^−1^ while maintaining a relatively high water and nitrogen use efficiency. The results of this experiment may differ from those of previous studies, owing to variations in climatic conditions, water and nitrogen supply levels, and crop types across agricultural regions. This indicates that the optimal water and nitrogen management strategies may exhibit regional specificity. Consequently, in practical application, model parameters should be appropriately adjusted to account for local conditions, thereby enabling the formulation of more precise water and nitrogen management strategies tailored to regional differences.

## 4. Materials and Methods

### 4.1. Experimental Site Profile

The experiment was conducted from April 2020 to October 2023 at Yimin Irrigation Experimental Station, Minle County, the middle part of the Hexi Corridor, Gansu Province (100°43′ E, 38°39′ N, 1970 m a.s.l.) (Figure 4). The climate in the experimental area is a temperate continental climate. The average annual precipitation is approximately 200 mm, the evaporation is 1900 mm, the average annual temperature is 6.0 °C, and the frost-free period lasts about 105 d. The experimental soil is Luvisols, with a maximum field water-holding capacity of 24% in the tillage layer and a bulk density of 1.46 g cm^−3^. The 0–20 cm soil layer has a pH of 7.5 and an organic matter content of 12.2 g kg^−1^. The available phosphorus, available potassium, and alkali-hydrolyzable nitrogen contents are 16.9 mg kg^−1^, 188.2 mg kg^−1^, and 72.9 mg kg^−1^, respectively, indicating moderate fertility. The precipitation and the average temperature over the experiments are shown in Figure 5.

### 4.2. Experimental Design

This study adopted a two-factor split-plot experimental design, with the main factor being the irrigation amount, which was precisely regulated, based on the reference crop evapotranspiration (ET_c_). Three gradient irrigation levels were set: sufficient water (100% ET_c_, W3), medium water (85% ET_c_, W2), and low water (70% ET_c_, W1). ET_C_ was determined by calculating the product of the reference crop evapotranspiration (ET_0_) and the crop coefficient (K_c_), using the Penman–Monteith formula (the K_c_ values are shown in Table 6). The secondary factor was the nitrogen application rate, which was determined based on the local agricultural extension department’s recommended nitrogen application rate. Three treatments were established: high nitrogen (330 kg·ha^−1^, N3), medium nitrogen (264 kg·ha^−1^, N2), and low nitrogen (198 kg·ha^−1^, N1), with no nitrogen application as the control treatment (0 kg·ha^−1^, N0). The experiment employed a completely randomized block design, resulting in a total of 12 treatment combinations. Each treatment was replicated three times, amounting to 36 experimental plots. Each plot measured 32 m^2^ (8 m × 4 m), with a 0.5 m isolation belt set between plots to prevent water and fertilizer interactions. The onion variety used in the experiment was “Crockett”, with a plant spacing of 0.2 m and a row spacing of 0.15 m, resulting in a planting density of 333,000 plants per hectare. All treatments were irrigated 9 times throughout the growth period, and the dates and times of irrigation are shown in Figure 6. Each treatment received 525 kg·ha^−1^ of diammonium phosphate, 150 kg·ha^−1^ of potassium sulphate, and 50% nitrogen fertilizer as basal fertilizer. During the leaf development stage, 50% nitrogen fertilizer was applied as a top dressing, and during the bulbification stage, 120 kg P ha^−1^ of phosphorus fertilizer and 150 kg K ha^−1^ of potassium fertilizer were applied as top dressings. The field application rates and normalized code values for each treatment are shown in Table 7.

### 4.3. Measurement Items and Methods

#### 4.3.1. Yield

During the onion maturity stage, each plot was harvested and weighed separately by using an electronic balance with a precision of 0.01 g. The average yield of the three replicates was taken as the actual yield for each treatment.

#### 4.3.2. Quality

During the onion maturity stage, five plants were randomly selected from each plot and transported to the laboratory for bulb quality analysis. The bulbs’ fresh weight was measured using an electronic balance with a precision of 0.01 g (Delixi Electric Co., Ltd., Wenzhou, Zhejiang, China), and bulbs’ transverse and longitudinal diameters were measured by using a vernier caliper with a precision of 0.01 mm (Delixi Electric Co., Ltd., Wenzhou, Zhejiang, China). Onion oil content was determined by steam distillation extraction, soluble sugars by the anthrone colorimetric method [46], soluble proteins by a Coomassie Brilliant Blue G-250 assay [47], pyruvate by the dinitrophenylhydrazine method [48], and vitamin C by the titration method [49].

#### 4.3.3. Economic Benefits

The formula for calculating farmland input costs is as follows [50]:(22)$$C_{ost}=\sum_{i=1}^{n}(C_{1}+C_{2}+\ldots +C_{n})$$(23)$$I_{ncome}=Y\times P$$(24)$$EP=C_{ost}-I_{ncome}$$
where *C_ost_* represents the total input cost (USD·ha^−1^); *C*_1_, *C*_2_, ..., *C_n_* represent the costs of various agricultural inputs in onion production (USD·ha^−1^); *I_ncome_* represents onion yield revenue, *Y* and *P* are the onion yield (kg·ha^−1^) and market unit price (USD·kg^−1^), respectively; and *EP* represents net profit.

#### 4.3.4. Soil Water Content

Soil water content was measured by the traditional drying and weighing method at a depth of 100 cm in five levels of 20, 40, 60, 80, and 100 cm. From transplanting to maturity, samples were taken every 10 d from the middle of two onion plants with a soil auger of 1.0 m in length and immediately encapsulated in an aluminum box and brought back to the laboratory for weighing (W1). Then, the lid of the aluminum box was uncovered, put to the bottom of the box, placed in an oven at 105 °C (provided by Qingdao Juchuang Jiaheng Analytical Instrument Co., Ltd.), and baked to a constant weight (about 8 h), and then moved into a desiccator and cooled to room temperature (about 20 min), then weighed (W2), and the soil moisture content was calculated.

The formula for calculating the soil water content is as follows:SWC (%) = (W1 − W2)/(W2 − W3) × 100%(25)
where SWC is the soil water content (%), W1 is the combined mass of fresh soil and the aluminum box (g), W2 is the combined mass of dry soil and the aluminum box (g), and W3 is the mass of the aluminum box (g).

The formula for soil water storage is as follows:SWS (mm) = h × ρ × ω × 10(26)
where SWS is the soil water storage (mm), h is the depth of the soil layer (cm), ρ is the soil bulk density (g•cm^−3^ ), and ω is the soil water content (%).

#### 4.3.5. Water–Nitrogen Use Efficiency

Crop water consumption (*ET*) and water–nitrogen use efficiency were calculated according to the method described by Pan et al. [51]. The formulas are as follows:(27)ET=F+I+ΔW−Q(28)F=αP(29)$$\Delta W=SWS_{B}-SWS_{A}$$
where *ET* represents the total water consumption of onion during the entire growth period (mm), *F* represents the amount of rainfall infiltration (mm), *I* represents the effective irrigation amount (mm), Δ*W* represents the change in soil water storage (mm), and *Q* represents recharge or seepage (mm). *P* represents the amount of rainfall (mm): when the rainfall is less than 5 mm, α is 0; when the rainfall is between 5 and 50 mm, α is 0.9; and when the rainfall is greater than 50 mm, α is 0.75. SWS_A_ represents the water storage capacity of the soil at a depth of 1.0 m after the onion harvest; SWS_B_ represents the water storage capacity of the soil at a depth of 1.0 m before onion transplanting.

Since the groundwater table in the experimental area is deeper than 10 m and the irrigation method is drip irrigation, there is no runoff drainage during the growth period, so groundwater recharge and seepage are neglected.

Water use efficiency (*WUE*, kg·m^−3^) was calculated as follows:(30)WUE=Y/ET
where *Y* represents the onion yield (kg·ha^−1^).

Irrigation water use efficiency (*IWUE*, kg·m^−3^) was calculated as follows:(31)IWUE=Y/I
where *I* represents the irrigation amount (mm).

Nitrogen partial factor productivity (*PFP_N_*, kg·kg^−1^) was calculated as follows:(32)$$PFP_{N}=Y/N$$
where *N* represents the nitrogen fertilizer application rate (kg).

Nitrogen agronomic use efficiency (*AUE_N_*, kg·kg^−1^) was calculated as follows:(33)$$AUE_{N}=(Y_{N}-Y_{0})/N$$
where *Y_N_* represents the onion yield in a nitrogen-fertilized plot (kg·ha^−1^) and *Y_0_* represents the onion yield in a zero-nitrogen control plot (kg·ha^−1^).

#### 4.3.6. Water–Nitrogen Economic Benefit Regression Model

The study used a water–nitrogen coupling model to construct a regression relationship between the irrigation amount, nitrogen application rate, and onion economic benefit. With the onion economic benefit as the dependent variable and the irrigation amount and nitrogen application rate as the independent variables, a water–nitrogen economic benefit regression model was established.

This model was expressed by using a bivariate quadratic regression equation, as follows:(34)$$y=a_{0}+a_{1}x_{1}+a_{2}x_{2}+a_{12}x_{1}x_{2}+a_{11}x_{1}^{2}+a_{22}x_{2}^{2}$$
where *y* represents the predicted economic benefits of onion. *a_0_* represents the constant term of the regression model. *a*_1_ and *a*_2_ represent the linear coefficient terms of the regression model. *a*_12_ represents the interaction coefficient term of the regression model. *a*_11_ and *a*_22_ represent the quadratic coefficients terms of the regression model.

### 4.4. Error Analysis

The simulated and measured values of economic benefits were evaluated using three metrics: mean relative error (MRE), coefficient of determination (R^2^), and root mean square error (RMSE). Significance analysis was performed with SPSS 17.0 software, and multiple comparisons were conducted using the Duncan analysis method, based on one-way ANOVA.(35)$$MRE=\frac{1}{n}\sum_{i=1}^{n}\frac{\left|S_{i}-M_{i}\right|}{S_{i}}\times 100\%$$(36)$$RMSE=\sqrt{\frac{1}{n}\sum_{i=1}^{n}{\left(S_{i}-M_{i}\right)}^{2}}$$(37)$$R^{2}=\frac{\sum_{i=1}^{n}\left(M_{i}-\bar{M})(S_{i}-\bar{S}\right)}{\sqrt{\sum_{i=1}^{n}\left(M_{i}-\bar{M}\right)}\sqrt{\sum_{i=1}^{n}\left(S_{i}-\bar{S}\right)}}$$
where *S_i_* represents the simulated value; *M_i_* represents the measured value; *i* represents the observation point; *n* represents the total number of observation points, with *n* being 12 for each treatment in this study; and $\bar{S}$ and $\bar{M}$ represent the average simulated value and the average measured value, respectively.

### 4.5. Data Statistics and Analysis

Data processing was performed using Microsoft Excel (Version 2010, Microsoft Corp., Raymond, WA, USA) software. The statistical analysis and regression model establishment were performed by using the LSD multiple comparison method in SPSS 22.0 (IBM, Inc., New York, NY, USA) software, while the graphs were plotted using Origin 8.0 (Origin Lab, Corp., Hampton, MA, USA).

## 5. Conclusions

In the Hexi Oasis region, under mulched drip irrigation, the coupling of irrigation and nitrogen application significantly impacts onion yield, quality, and water–nitrogen use efficiency. Onion quality accumulation is closely linked to water and nitrogen management. Four years of experiments indicate that sufficient water (100% ET_c_) and medium nitrogen (264 kg·ha^−1^) most effectively promoted the bulb fresh weight, diameter, diameter, length, and soluble sugar formation. Medium water (85% ET_c_) and medium nitrogen (264 kg·ha^−1^) was more conducive to the accumulation of onion oil, soluble protein, vitamin C, and propionic acid. Increasing both the irrigation amount and the nitrogen application rate enhanced the onion yield and economic benefits. However, when the nitrogen application rate exceeded 266 kg·ha^−1^, the yield-increasing effect diminished significantly, while simultaneously raising the farmland input costs and reducing the net profits. Both excessive and insufficient irrigation and nitrogen application limited improvements in the onion’s WUE, IWUE, PFP_N_, and AUE_N_. By establishing a regression model equation for water–nitrogen-economic benefits, it was found to correlate highly with actual economic benefits, enabling the prediction of optimal water–nitrogen allocation schemes across different target economic benefit ranges. Taking into account the growing environment of onions in the oasis agricultural areas of the Hexi Corridor and the effects of varying water and nitrogen supplies on onion’s quality, yield, economic benefits, and water–nitrogen use efficiency, it was determined that with an irrigation amount at 100% ET_c_ and nitrogen application rates of 266 kg·ha^−1^, higher economic benefits and water–nitrogen use efficiency can be achieved, alongside superior bulb quality. This strategy provides decision-making guidance for enhancing income and quality within the onion industry of the Hexi Corridor oasis agricultural areas. Future research should integrate environmental indicators such as N_2_O emissions, nitrogen leaching, and carbon footprint to provide a more comprehensive assessment of sustainable onion production.

## Figures and Tables

**Figure 1 plants-15-00006-f001:**
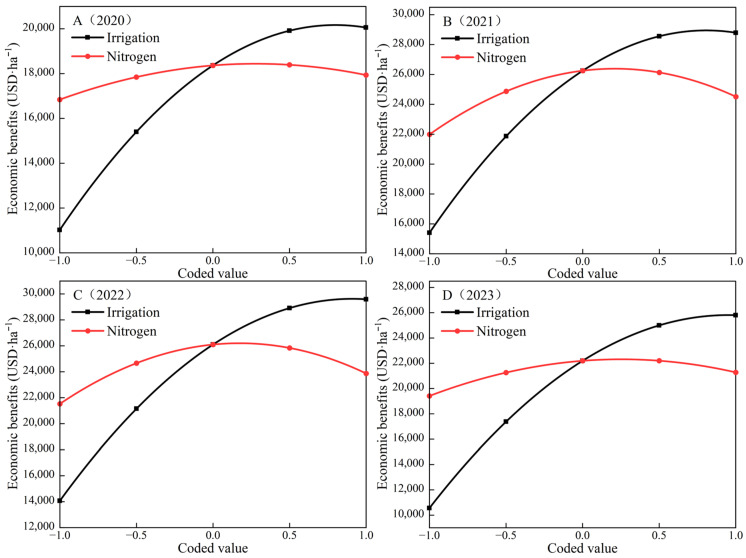
Effect curve of single-factor effects on the economic benefits of onion.

**Figure 2 plants-15-00006-f002:**
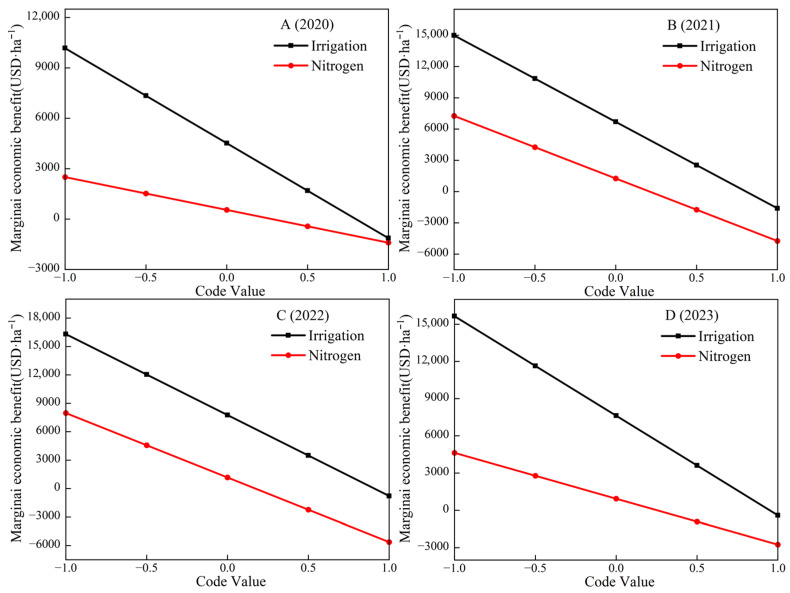
Effect curve of single-factor effects on the marginal economic benefit of onion.

**Figure 3 plants-15-00006-f003:**
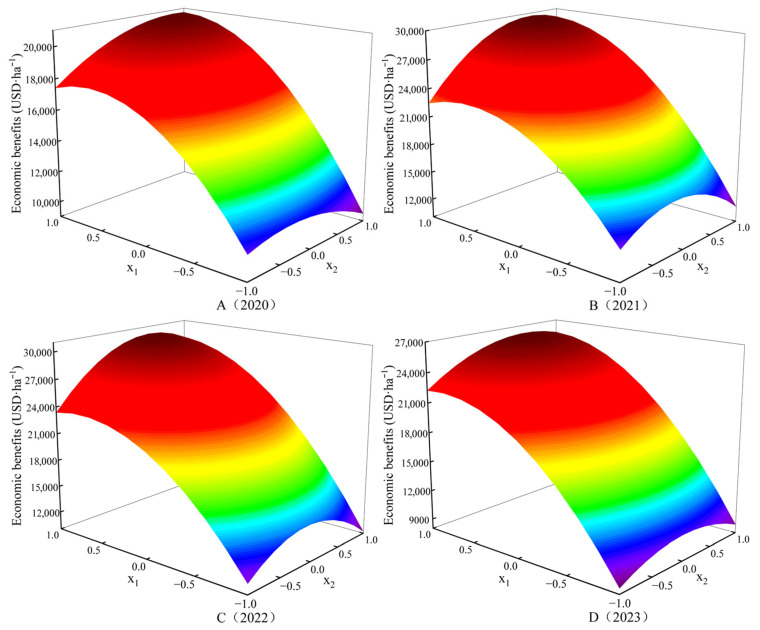
Effects of water–nitrogen interaction on the economic benefits of onion.

**Figure 4 plants-15-00006-f004:**
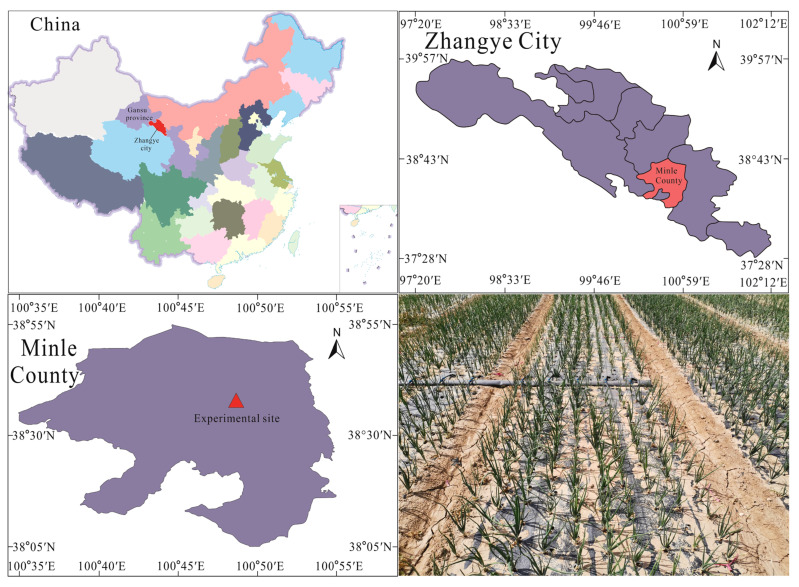
Location of the experimental site.

**Figure 5 plants-15-00006-f005:**
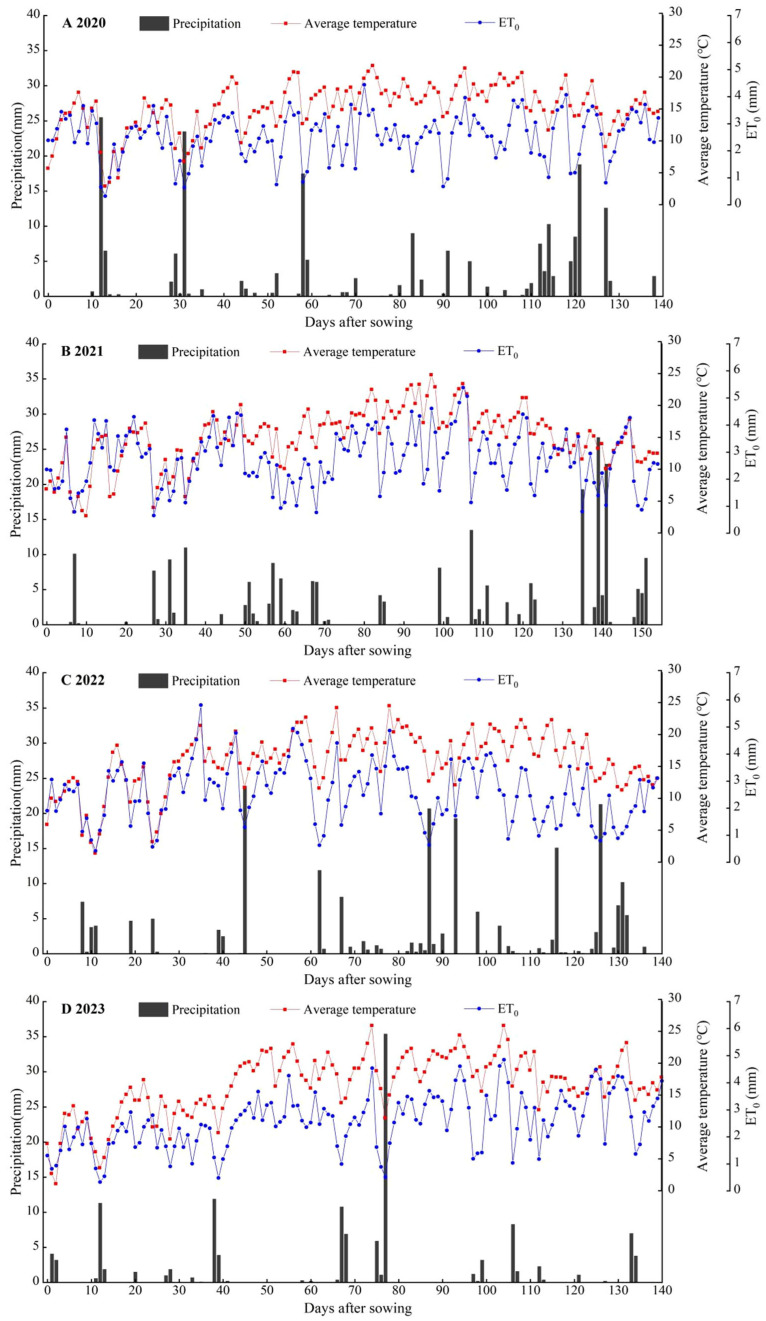
Daily variation in average temperature, reference crop evapotranspiration (ET_0_), and precipitation throughout onion growing seasons of 2020 (**A**), 2021 (**B**), 2022 (**C**), and 2023 (**D**).

**Figure 6 plants-15-00006-f006:**
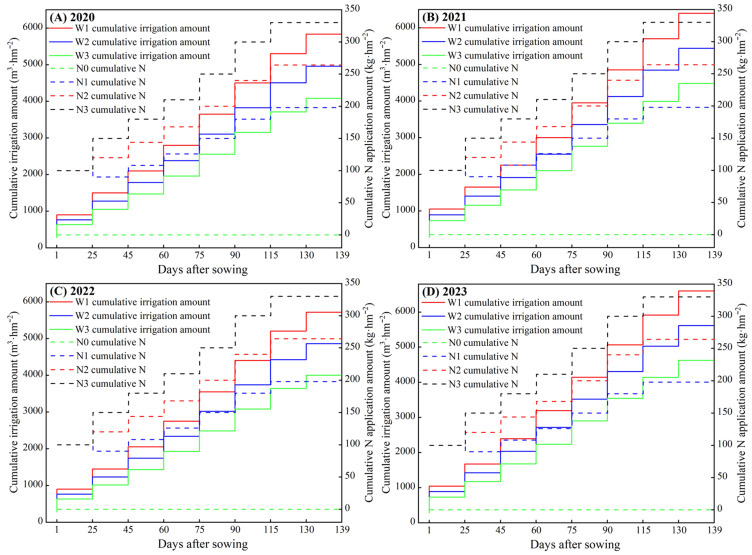
Design of different irrigation systems of onion in 2020, 2021, 2022, and 2023.

**Table 1 plants-15-00006-t001:** The effect of water–nitrogen interaction on onion quality.

Year	Treatment	Bulb Fresh Weight (g)	Diameter (mm)	Length (mm)	Onion Oil (%)	Soluble Sugars (mg·g^−1^)	Soluble Proteins (mg·g^−1^)	Vitamin C (mg·100g^−1^)	Propionic Acid (mg·g^−1^)
2020	N0W1	164.65 ± 4.06 i	58.91 ± 2.40 g	61.84 ± 1.09 g	0.29 ± 0.011 h	142.05 ± 7.58 fg	16.44 ± 0.38 h	15.58 ± 0.45 g	0.20 ± 0.009 h
	N0W2	239.04 ± 9.12 f	65.40 ± 3.19 f	77.98 ± 2.46 f	0.35 ± 0.015 def	165.30 ± 6.21 cd	18.48 ± 0.68 fg	16.51 ± 0.53 ef	0.25 ± 0.006 ef
	N0W3	262.19 ± 4.28 e	79.45 ± 3.00 de	85.23 ± 4.51 de	0.34 ± 0.011 ef	166.27 ± 3.30 cd	17.75 ± 0.49 g	17.91 ± 0.05 cd	0.23 ± 0.003 g
	N1W1	227.46 ± 11.29 fg	75.47 ± 1.76 e	83.22 ± 1.94 de	0.30 ± 0.015 h	147.48 ± 5.70 f	19.36 ± 1.12 ef	16.05 ± 0.82 fg	0.21 ± 0.011 h
	N1W2	325.12 ± 16.76 d	82.33 ± 3.15 cd	85.53 ± 4.20 de	0.37 ± 0.020 cd	172.35 ± 2.63 bc	21.26 ± 0.72 bc	18.28 ± 0.61 bc	0.27 ± 0.005 cd
	N1W3	352.59 ± 13.38 c	85.88 ± 4.52 bc	87.14 ± 3.07 d	0.36 ± 0.019 de	178.69 ± 5.345 ab	18.81 ± 0.65 fg	18.84 ± 0.11 b	0.26 ± 0.013 de
	N2W1	213.99 ± 11.21 g	74.30 ± 2.37 e	80.55 ± 1.09 ef	0.33 ± 0.016 fg	151.26 ± 2.75 ef	22.90 ± 0.34 a	17.59 ± 0.35 cd	0.24 ± 0.008 fg
	N2W2	381.11 ± 10.59 b	90.43 ± 4.31 ab	99.84 ± 1.85 b	0.42 ± 0.013 a	180.45 ± 7.64 ab	22.98 ± 0.20 a	20.70 ± 0.74 a	0.30 ± 0.012 a
	N2W3	409.12 ± 23.34 a	90.97 ± 3.33 ab	102.71 ± 4.26 b	0.42 ± 0.021 a	184.05 ± 4.30 a	20.61 ± 0.61 cd	20.43 ± 0.40 a	0.29 ± 0.014 ab
	N3W1	187.69 ± 7.53 h	69.00 ± 2.03 f	78.01 ± 1.95 f	0.31 ± 0.012 gh	136.71 ± 5.98 g	21.95 ± 0.92 ab	17.16 ± 0.20 de	0.23 ± 0.010 g
	N3W2	371.33 ± 11.86 bc	87.65 ± 2.01 b	93.86 ± 1.73 c	0.40 ± 0.008 ab	157.96 ± 6.58 de	22.77 ± 0.53 a	20.05 ± 0.74 a	0.30 ± 0.011 a
	N3W3	412.54 ± 18.63 a	93.25 ± 1.38 a	109.18 ± 3.02 a	0.39 ± 0.011 bc	160.30 ± 7.94 de	20.02 ± 0.34 de	19.77 ± 0.46 a	0.28 ± 0.012 bc
Significance (F)	W	487.71 ***	116.30 ***	156.13 ***	101.10 ***	83.69 ***	32.47 ***	93.77 ***	116.98 ***
	N	130.14 ***	63.96 ***	90.99 ***	31.66 ***	21.51 ***	96.20 ***	57.94 ***	46.99 ***
	W × N	17.75 ***	6.24 ***	14.13 ***	1.35 ns	0.59 ns	5.43 **	3.15 *	1.00 ns
2021	N0W1	142.67 ± 6.20 h	54.68 ± 1.47 h	66.33 ± 2.78 e	0.26 ± 0.011 f	133.86 ± 0.73 ij	16.09 ± 0.94 g	15.32 ± 0.67 g	0.18 ± 0.003 h
	N0W2	198.57 ± 7.74 ef	70.95 ± 1.83 ef	72.80 ± 1.00 d	0.32 ± 0.006 cd	154.32 ± 4.89 ef	18.18 ± 0.95 ef	16.49 ± 0.16 ef	0.24 ± 0.013 de
	N0W3	217.56 ± 10.61 e	73.92 ± 3.63 de	73.24 ± 2.96 d	0.31 ± 0.016 d	159.17 ± 2.66 de	17.62 ± 0.41 f	17.53 ± 0.24 de	0.22 ± 0.011 fg
	N1W1	189.37 ± 11.07 fg	68.81 ± 1.87 fg	71.99 ± 3.19 d	0.27 ± 0.004 f	137.46 ± 7.88 hi	19.29 ± 0.29 de	16.08 ± 0.87 fg	0.19 ± 0.007 h
	N1W2	297.93 ± 13.58 d	75.53 ± 2.03 d	83.71 ± 4.55 bc	0.35 ± 0.018 b	165.03 ± 6.32 cd	21.05 ± 1.02 bc	18.27 ± 0.89 cd	0.25 ± 0.013 d
	N1W3	307.94 ± 10.69 cd	76.44 ± 2.14 cd	83.69 ± 2.48 bc	0.34 ± 0.008 bc	170.18 ± 4.70 bc	18.55 ± 0.35 ef	18.91 ± 0.30 bc	0.24 ± 0.008 de
	N2W1	192.49 ± 6.71 fg	64.79 ± 1.16 g	66.52 ± 2.16 e	0.30 ± 0.016 de	140.91 ± 1.09 hi	22.02 ± 0.14 ab	17.38 ± 0.80 de	0.23 ± 0.010 ef
	N2W2	367.93 ± 16.83 b	80.00 ± 2.67 c	87.51 ± 3.12 b	0.40 ± 0.013 a	176.34 ± 6.48 ab	22.51 ± 0.71 a	20.62 ± 0.42 a	0.31 ± 0.015 a
	N2W3	406.18 ± 8.38 a	91.26 ± 4.22 a	98.24 ± 2.32 a	0.39 ± 0.020 a	181.52 ± 3.68 a	20.74 ± 1.08 bc	20.41 ± 0.64 a	0.29 ± 0.010 b
	N3W1	174.48 ± 5.03 g	67.82 ± 2.16 fg	63.67 ± 3.03 e	0.28 ± 0.013 ef	127.70 ± 1.55 j	21.46 ± 0.32 ab	17.05 ± 0.78 ef	0.21 ± 0.009 g
	N3W2	322.56 ± 8.10 c	77.88 ± 1.66 cd	81.23 ± 1.71 c	0.38 ± 0.014 a	144.62 ± 7.00 gh	21.93 ± 0.68 ab	19.86 ± 0.71 ab	0.30 ± 0.015 ab
	N3W3	399.12 ± 20.57 a	87.06 ± 1.56 b	97.80 ± 3.41 a	0.38 ± 0.019 a	149.47 ± 1.29 fg	19.81 ± 0.66 cd	19.53 ± 0.58 ab	0.27 ± 0.010 c
Significance (F)	W	638.91 ***	183.28 ***	170.15 ***	135.32 ***	141.15 ***	19.30 ***	62.54 ***	140.16 ***
	N	246.34 ***	48.85 ***	35.80 ***	39.55 ***	49.49 ***	71.01 ***	39.08 ***	64.08 ***
	W × N	34.58 ***	9.41 ***	18.22 ***	2.25 ns	3.10 *	3.51 *	1.72 ns	1.60 ns
2022	N0W1	113.28 ± 7.21 h	56.34 ± 2.71 e	52.23 ± 2.98 f	0.24 ± 0.013 h	129.62 ± 4.96 gh	16.12 ± 0.76 e	15.27 ± 0.49 h	0.16 ± 0.003 g
	N0W2	141.82 ± 9.17 g	59.04 ± 3.04 e	69.82 ± 3.68 de	0.31 ± 0.013 ef	150.46 ± 4.53 de	17.81 ± 0.73 d	16.18 ± 0.41 gh	0.22 ± 0.007 de
	N0W3	207.42 ± 10.16 e	66.82 ± 2.97 d	75.15 ± 3.83 cd	0.30 ± 0.011 f	160.01 ± 6.49 cd	17.46 ± 0.85 de	17.49 ± 0.59 ef	0.21 ± 0.003 e
	N1W1	149.82 ± 9.44 fg	64.53 ± 2.72 d	70.15 ± 3.12 de	0.25 ± 0.005 gh	134.38 ± 1.08 fg	19.05 ± 0.58 cd	15.93 ± 0.60 gh	0.18 ± 0.007 f
	N1W2	257.75 ± 7.73 d	72.38 ± 3.77 c	76.82 ± 3.92 c	0.35 ± 0.008 cd	164.72 ± 7.58 c	20.94 ± 0.70 ab	18.05 ± 0.69 de	0.24 ± 0.010 c
	N1W3	297.89 ± 12.51 c	74.23 ± 3.09 c	77.77 ± 3.67 c	0.33 ± 0.011 de	168.28 ± 3.97 bc	18.32 ± 1.00 d	18.64 ± 0.37 cd	0.23 ± 0.009 cd
	N2W1	160.21 ± 9.85 f	66.71 ± 1.37 d	71.11 ± 1.86 de	0.30 ± 0.013 f	136.51 ± 6.79 fg	22.08 ± 1.08 a	17.02 ± 0.71 efg	0.21 ± 0.009 e
	N2W2	342.93 ± 15.84 b	84.54 ± 3.44 b	83.52 ± 2.62 b	0.41 ± 0.011 a	175.39 ± 8.84 ab	22.32 ± 0.40 a	20.65 ± 0.87 a	0.29 ± 0.007 a
	N2W3	374.50 ± 7.77 a	95.97 ± 3.79 a	96.58 ± 1.92 a	0.39 ± 0.015 ab	182.43 ± 9.25 a	20.65 ± 1.46 ab	20.27 ± 0.86 ab	0.28 ± 0.013 ab
	N3W1	133.83 ± 6.01 g	58.70 ± 3.10 e	67.21 ± 2.79 e	0.27 ± 0.016 g	124.08 ± 0.68 h	21.29 ± 1.19 ab	16.85 ± 0.70 fg	0.19 ± 0.006 f
	N3W2	298.46 ± 14.93 c	77.08 ± 3.47 c	77.28 ± 2.96 c	0.38 ± 0.012 b	141.50 ± 5.77 ef	21.74 ± 0.43 a	19.51 ± 0.66 bc	0.29 ± 0.008 a
	N3W3	365.91 ± 7.81 a	87.98 ± 2.30 b	85.74 ± 3.88 b	0.37 ± 0.012 bc	144.92 ± 1.00 ef	19.93 ± 0.96 bc	19.28 ± 0.55 bc	0.27 ± 0.010 b
Significance (F)	W	883.95 ***	126.33 ***	105.27 ***	230.22 ***	108.15 ***	10.02**	61.13 ***	280.72 ***
	N	305.26 ***	78.84 ***	48.94 ***	81.90 ***	38.29 ***	45.50 ***	36.93 ***	112.41 ***
	W × N	43.12 ***	10.82 ***	5.40 **	2.38 ns	3.10 *	2.03 ns	2.55 *	4.47 **
2023	N0W1	86.35 ± 4.02 h	54.26 ± 2.87 g	58.54 ± 2.75 e	0.22 ± 0.004 g	125.65 ± 4.43 gh	15.98 ± 0.37 i	15.04 ± 0.43 h	0.14 ± 0.005 h
	N0W2	159.54 ± 7.79 f	59.03 ± 1.29 fg	64.28 ± 2.65 d	0.29 ± 0.008 e	147.38 ± 1.89 de	17.57 ± 0.65 gh	15.75 ± 0.51 gh	0.21 ± 0.011 de
	N0W3	227.52 ± 8.60 e	70.55 ± 4.83 d	72.43 ± 3.54 c	0.28 ± 0.009 e	154.90 ± 9.17 cd	17.02 ± 0.47 h	17.18 ± 0.38 ef	0.19 ± 0.005 f
	N1W1	140.71 ± 5.25 g	65.43 ± 1.64 de	64.56 ± 1.97 d	0.23 ± 0.011 g	128.64 ± 7.25 gh	18.93 ± 1.10 ef	15.63 ± 0.80 gh	0.16 ± 0.007 g
	N1W2	286.34 ± 10.97 d	77.78 ± 2.56 c	82.36 ± 2.75 b	0.35 ± 0.009 c	159.86 ± 9.60 bc	20.75 ± 0.70 bcd	17.85 ± 0.59 de	0.25 ± 0.011 c
	N1W3	336.61 ± 17.71 bc	85.36 ± 2.48 ab	93.90 ± 5.12 a	0.31 ± 0.013 d	164.75 ± 8.45 bc	18.40 ± 0.64 fg	18.41 ± 0.50 cd	0.22 ± 0.010 d
	N2W1	146.08 ± 7.43 fg	67.93 ± 3.40 de	71.86 ± 3.52 c	0.27 ± 0.012 e	132.83 ± 6.28 fg	21.85 ± 0.33 ab	16.74 ± 0.33 f	0.20 ± 0.009 ef
	N2W2	347.38 ± 16.55 b	86.02 ± 2.37 ab	93.13 ± 1.69 a	0.40 ± 0.018 a	170.21 ± 8.62 ab	22.17 ± 0.19 a	20.49 ± 0.73 a	0.29 ± 0.007 a
	N2W3	400.32 ± 14.64 a	89.14 ± 3.94 a	98.33 ± 2.46 a	0.38 ± 0.020 b	179.37 ± 3.57 a	20.29 ± 0.60 cd	20.07 ± 0.39 ab	0.27 ± 0.012 b
	N3W1	136.02 ± 6.23 g	63.85 ± 2.52 ef	62.71 ± 3.52 de	0.25 ± 0.001 f	119.82 ± 3.04 h	21.03 ± 0.88 bc	16.28 ± 0.56 fg	0.17 ± 0.006 g
	N3W2	327.26 ± 7.50 c	82.56 ± 3.68 bc	87.02 ± 1.73 b	0.36 ± 0.015 bc	137.09 ± 5.53 efg	21.48 ± 0.50 ab	19.36 ± 0.71 bc	0.26 ± 0.009 bc
	N3W3	389.33 ± 5.76 a	87.60 ± 2.74 ab	96.54 ± 4.66 a	0.36 ± 0.009 bc	141.56 ± 6.11 ef	19.67 ± 0.34 de	18.92 ± 0.44 c	0.26 ± 0.009 bc
Significance (F)	W	1333.80 ***	141.23 ***	203.39 ***	286.85 ***	85.71 ***	22.14 ***	88.75 ***	317.06 ***
	N	335.95 ***	76.75 ***	82.44 ***	88.85 ***	29.17 ***	97.39 ***	52.19 ***	114.10 ***
	W × N	29.89 ***	3.57 *	7.14 ***	5.69 **	2.26 ns	3.43 *	4.56 **	3.61 *

Note: Different letters following the same column of values indicate differences at the *p* < 0.05 level. * indicates a significant difference at the *p* < 0.05 level, ** indicates a significant difference at the *p* < 0.01 level, *** indicates a significant difference at the *p* < 0.001 level. The ns means not significant at the level of p ≥ 0.05. W indicates irrigation, and N indicates nitrogen fertilizer.

**Table 2 plants-15-00006-t002:** The effect of the water–nitrogen interaction on the onion yield.

Treatment	2020 Year (t·ha^−1^)	2021 Year (t·ha^−1^)	2022 Year (t·ha^−1^)	2023 Year (t·ha^−1^)
N0W1	55.25 ± 1.22 i	46.77 ± 0.43 h	36.98 ± 0.87 i	28.26 ± 0.86 h
N0W2	79.15 ± 1.38 f	65.37 ± 0.79 ef	46.88 ± 1.32 gh	53.38 ± 0.67 f
N0W3	88.06 ± 2.06 e	70.75 ± 2.06 e	67.97 ± 1.47 e	74.76 ± 1.14 e
N1W1	75.44 ± 1.97 fg	62.91 ± 1.86 f	51.27 ± 0.69 fg	46.62 ± 0.68 g
N1W2	106.72 ± 2.84 d	99.72 ± 0.83 d	86.13 ± 1.91 d	94.71 ± 1.79 d
N1W3	118.78 ± 1.62 c	100.59 ± 2.03 d	99.29 ± 3.01 c	113.30 ± 1.89 bc
N2W1	69.71 ± 2.00 gh	63.95 ± 0.91 f	53.02 ± 0.98 f	49.43 ± 1.43 fg
N2W2	127.06 ± 3.70 b	123.19 ± 2.64 b	115.63 ± 3.18 b	115.23 ± 1.83 b
N2W3	135.44 ± 3.96 a	134.41 ± 2.61 a	125.94 ± 2.66 a	132.26 ± 3.37 a
N3W1	63.88 ± 1.84 h	56.75 ± 1.02 g	45.22 ± 0.77 h	45.86 ± 1.04 g
N3W2	123.30 ± 2.90 bc	109.15 ± 1.60 c	100.19 ± 1.32 c	108.95 ± 2.80 c
N3W3	136.93 ± 2.69 a	131.84 ± 3.68 a	120.89 ± 1.04 ab	128.25 ± 0.90 a
W	520.40 ***	803.88 ***	1044.05 ***	1710.28 ***
N	134.63 ***	325.68 ***	386.08 ***	442.83 ***
W × N	18.41 ***	46.02 ***	52.59 ***	35.50 ***

Note: Different letters following the same column of values indicate differences at the *p* < 0.05 level. *** indicates a significant difference at the *p* < 0.001 level, W indicates irrigation, and N indicates nitrogen fertilizer.

**Table 3 plants-15-00006-t003:** The effect of the water–nitrogen interaction on the onions’ water–nitrogen use efficiency.

Year	Treatment	ET (mm)	WUE (kg·m^−3^)	IWUE (kg·m^−3^)	PFP_N_ (kg·kg^−1^)	AUE_N_ (kg·kg^−1^)
2020	N0W1	557.91 ± 21.55 c	9.91 ± 0.38 h	13.53 ± 0.46 h	—	—
	N0W2	568.44 ± 19.71 c	13.93 ± 0.39 d	15.96 ± 0.37 g	—	—
	N0W3	671.34 ± 28.30 b	13.13 ± 0.54 e	15.10 ± 0.20 g	—	—
	N1W1	570.50 ± 29.71 c	13.23 ± 0.23 de	18.48 ± 0.73 e	381.03 ± 15.09 f	101.99 ± 8.71 e
	N1W2	679.27 ± 13.62 b	15.71 ± 0.13 c	21.52 ± 0.67 c	538.99 ± 9.16 b	139.25 ± 7.35 cd
	N1W3	775.91 ± 15.78 a	15.31 ± 0.14 c	20.36 ± 0.50 d	599.91 ± 14.71 a	155.15 ± 8.85 b
	N2W1	576.52 ± 17.89 c	12.09 ± 0.23 f	17.07 ± 0.60 f	264.05 ± 9.34 g	54.77 ± 2.60 f
	N2W2	682.41 ± 25.13 b	18.63 ± 0.39 a	25.63 ± 0.61 a	481.29 ± 11.45 d	181.49 ± 4.81 a
	N2W3	777.53 ± 26.88 a	17.43 ± 0.30 b	23.22 ± 0.45 b	513.05 ± 9.86 c	179.47 ± 5.48 a
	N3W1	581.48 ± 33.58 c	11.00 ± 0.35 g	15.65 ± 0.67 g	193.59 ± 8.28 h	26.17 ± 5.12 g
	N3W2	685.60 ± 29.55 b	18.01 ± 1.04 ab	24.87 ± 0.54 a	373.62 ± 8.05 f	133.78 ± 3.41 d
	N3W3	786.51 ± 31.48 a	17.42 ± 0.38 b	23.47 ± 0.64 b	414.93 ± 11.32 e	148.07 ± 8.20 bc
Significance (F)	W	155.24 ***	451.81 ***	389.32 ***	1088.16 ***	660.54 ***
	N	23.00 ***	124.44 ***	333.14 ***	589.30 ***	79.48 ***
	W × N	3.42 *	22.09 ***	40.60 ***	5.70 **	47.73 ***
2021	N0W1	566.17 ± 23.10 d	8.26 ± 0.15 i	10.44 ± 0.39 g	—	—
	N0W2	617.88 ± 30.18 c	10.58 ± 0.35 fg	12.02 ± 0.25 f	—	—
	N0W3	716.23 ± 27.06 b	9.88 ± 0.33 g	11.06 ± 0.56 g	—	—
	N1W1	584.92 ± 13.65 cd	10.75 ± 0.19 f	14.05 ± 0.52 e	317.72 ± 11.83 d	81.52 ± 4.98 f
	N1W2	703.48 ± 26.94 b	14.17 ± 0.40 d	18.34 ± 0.27 c	503.61 ± 7.31 a	173.48 ± 4.56 c
	N1W3	848.86 ± 44.66 a	11.85 ± 0.47 e	15.72 ± 0.55 d	508.05 ± 17.76 a	150.75 ± 1.29 d
	N2W1	607.66 ± 19.41 cd	10.52 ± 0.29 fg	14.28 ± 0.35 e	242.25 ± 5.98 e	65.11 ± 5.99 g
	N2W2	731.76 ± 34.86 b	16.84 ± 0.17 a	22.65 ± 0.84 a	466.64 ± 17.30 b	219.0 ± 4 14.15 b
	N2W3	850.54 ± 31.10 a	15.80 ± 0.79 b	21.01 ± 0.70 b	509.13 ± 17.08 a	241.15 ± 10.24 a
	N3W1	628.71 ± 18.50 c	9.03 ± 0.22 h	12.67 ± 0.39 f	171.95 ± 5.33 f	30.24 ± 8.00 h
	N3W2	733.67 ± 16.80 b	14.88 ± 0.27 cd	20.07 ± 0.51 b	330.76 ± 8.40 d	132.68 ± 5.75 e
	N3W3	855.11 ± 12.66 a	15.42 ± 0.90 bc	20.61 ± 1.00 b	399.51 ± 19.35 c	185.13 ± 12.09 c
Significance (F)	W	208.92 ***	345.96 ***	298.59 ***	751.50 ***	674.80 ***
	N	29.74 ***	191.17 ***	344.95 ***	274.62 ***	116.46 ***
	W × N	2.80 *	30.63 ***	37.63 ***	10.06 ***	40.83 ***
2022	N0W1	485.30 ± 14.15 e	7.62 ± 0.32 h	9.24 ± 0.43 h	—	—
	N0W2	540.23 ± 27.12 cd	8.68 ± 0.12 g	9.65 ± 0.40 h	—	—
	N0W3	608.05 ± 25.48 b	11.18 ± 0.38 e	11.89 ± 0.58 fg	—	—
	N1W1	512.23 ± 26.97 de	10.01 ± 0.23 f	12.81 ± 0.39 ef	258.93 ± 7.88 f	72.15 ± 3.17 e
	N1W2	603.72 ± 16.64 b	14.27 ± 0.12 d	17.72 ± 0.59 d	434.99 ± 14.54 c	198.21 ± 8.21 c
	N1W3	708.51 ± 26.58 a	14.01 ± 0.19 d	17.37 ± 0.89 d	501.45 ± 25.72 a	158.15 ± 9.06 d
	N2W1	540.29 ± 16.02 cd	9.81 ± 0.27 f	13.25 ± 0.65 e	200.82 ± 9.82 g	60.73 ± 5.49 f
	N2W2	609.14 ± 26.24 b	18.98 ± 0.49 a	23.79 ± 0.43 a	438.00 ± 7.4 c	260.41 ± 6.08 a
	N2W3	709.64 ± 20.16 a	17.75 ± 0.34 b	22.03 ± 0.55 b	477.06 ± 11.91 b	219.59 ± 1.55 b
	N3W1	557.40 ± 18.14 c	8.11 ± 0.53 gh	11.30 ± 0.63 g	137.03 ± 7.69 h	24.97 ± 3.29 g
	N3W2	610.13 ± 22.27 b	16.42 ± 0.41 c	20.62 ± 0.41 c	303.60 ± 6.05 e	161.53 ± 4.42 d
	N3W3	711.58 ± 32.93 a	16.99 ± 0.44 c	21.14 ± 0.67 bc	366.32 ± 11.64 d	160.34 ± 3.57 d
Significance (F)	W	142.96 ***	1177.77 ***	499.26 ***	944.29 ***	2011.70 ***
	N	23.20 ***	550.25 ***	454.53 ***	258.15 ***	313.36 ***
	W × N	1.43 ns	101.54 ***	54.84 ***	6.98 **	57.57 ***
2023	N0W1	529.29 ± 25.22 h	5.34 ± 0.24 h	6.11 ± 0.24 g	—	—
	N0W2	598.45 ± 20.83 fg	8.92 ± 0.16 f	9.51 ± 0.23 f	—	—
	N0W3	673.16 ± 26.78 de	11.11 ± 0.15 e	11.32 ± 0.60 d	—	—
	N1W1	562.83 ± 24.64 gh	8.28 ± 0.16 f	10.08 ± 0.34 ef	235.43 ± 7.91 g	92.71 ± 270 e
	N1W2	662.33 ± 17.50 e	14.30 ± 0.32 d	16.87 ± 0.33 c	478.35 ± 9.43 c	208.78 ± 2.97 b
	N1W3	761.36 ± 13.20 ab	14.88 ± 0.20 cd	17.16 ± 0.41 c	572.22 ± 13.72 a	194.64 ± 8.02 c
	N2W1	575.27 ± 26.25 gh	8.59 ± 0.23 f	10.69 ± 0.53 de	187.23 ± 9.36 h	80.18 ± 5.40 e
	N2W2	712.31 ± 28.65 cd	16.18 ± 0.16 b	20.53 ± 0.81 a	436.49 ± 17.16 d	234.31 ± 12.83 a
	N2W3	773.92 ± 32.39 ab	17.09 ± 0.62 a	20.03 ± 0.40 ab	501.00 ± 9.90 b	217.82 ± 8.51 b
	N3W1	630.60 ± 34.07 ef	7.27 ± 0.20 g	9.92 ± 0.40 ef	138.97 ± 5.54 i	53.33 ± 2.24 f
	N3W2	727.18 ± 35.55 bc	14.98 ± 0.64 c	19.41 ± 0.52 b	330.14 ± 8.92 f	168.40 ± 7.34 d
	N3W3	789.68 ± 35.95 a	16.24 ± 0.67 b	19.42 ± 0.85 b	388.63 ± 17.03 e	162.08 ± 8.71 d
Significance (F)	W	121.42 ***	1406.70 ***	885.96 ***	1628.09 ***	844.64 ***
	N	28.56 ***	381.24 ***	461.81 ***	345.08 ***	112.09 ***
	W × N	1.22 ns	23.96 ***	27.05 ***	12.58 ***	8.26 **

Note: Different letters following the same column of values indicate differences at the *p* < 0.05 level. * indicates a significant difference at the *p* < 0.05 level, ** indicates a significant difference at the *p* < 0.01 level, *** indicates a significant difference at the *p* < 0.001 level. The ns means not significant at the level of p ≥ 0.05. W indicates irrigation, and N indicates nitrogen fertilizer.

**Table 4 plants-15-00006-t004:** The effect of water–nitrogen interaction on onion economic benefits.

Year	Treatment	Water Input (USD·ha^−1^)	Fertilizer Input (USD·ha^−1^)	Other Inputs (USD·ha^−1^)	Total Input (USD·ha^−1^)	Economic Benefit (USD·ha^−1^)	Net Profit (USD·ha^−1^)
2020	N0W1	133.7	797.5	4821.4	5752.6	8221.8 ± 181.58 i	2469.2 ± 181.58 i
	N0W2	162.3	797.5	4821.4	5781.2	11,777.9 ± 204.61 f	5996.7 ± 204.61 f
	N0W3	191.0	797.5	4821.4	5809.9	13,104.8 ± 306.71 e	7294.9 ± 306.71 e
	N1W1	133.7	944.9	4821.4	5900.0	11,226.8 ± 293.00 fg	5326.8 ± 293.00 fg
	N1W2	162.3	944.9	4821.4	5928.6	15,880.9 ± 422.31 d	9952.3 ± 422.31 d
	N1W3	191.0	944.9	4821.4	5957.3	17,676.1 ± 240.88 c	11,718.8 ± 240.88 c
	N2W1	133.7	994.0	4821.4	5949.1	10,373.4 ± 298.10 gh	4424.3 ± 298.10 gh
	N2W2	162.3	994.0	4821.4	5977.7	18,907.9 ± 549.86 b	12,930.2 ± 549.86 b
	N2W3	191.0	994.0	4821.4	6006.4	20,155.4 ± 589.51 a	14,149.0 ± 589.51 a
	N3W1	133.7	1043.1	4821.4	5998.2	9506.7 ± 273.23 h	3508.5 ± 273.33 h
	N3W2	162.3	1043.1	4821.4	6026.8	18,347.6 ± 431.32 bc	12,320.8 ± 431.32 bc
	N3W3	191.0	1043.1	4821.4	6055.5	20,376.3 ± 400.85 a	14,320.8 ± 400.85 a
Significance (F)	W	—	—	—	—	520.40 ***	513.74 ***
	N	—	—	—	—	134.63 ***	123.80 ***
	W × N	—	—	—	—	18.41 ***	18.41 ***
2021	N0W1	154.2	824.7	5539.9	6518.8	10,246.1 ± 94.06 h	3727.3 ± 94.06 h
	N0W2	187.2	824.7	5539.9	6551.8	14,321.1 ± 172.31 ef	7769.3 ± 172.31 ef
	N0W3	220.3	824.7	5539.9	6584.9	15,500.0 ± 421.15 e	8915.1 ± 451.15 e
	N1W1	154.2	986.4	5539.9	6680.5	13,782.6 ± 407.60 f	7102.1 ± 407.60 f
	N1W2	187.2	986.4	5539.9	6713.5	21,846.9 ± 182.59 d	15,133.4 ± 182.59 d
	N1W3	220.3	986.4	5539.9	6746.6	22,039.5 ± 444.50 d	15,292.9 ± 444.50 d
	N2W1	154.2	1040.3	5539.9	6734.4	14,011.9 ± 199.23 f	7277.5 ± 199.23 f
	N2W2	187.2	1040.3	5539.9	6767.4	26,990.5 ± 578.01 b	20,223.1 ± 578.01 b
	N2W3	220.3	1040.3	5539.9	6800.5	29,448.3 ± 570.86 a	22,647.8 ± 570.86 a
	N3W1	154.2	1094.2	5539.9	6788.3	12,432.4 ± 222.83 g	5644.1 ± 222.83 g
	N3W2	187.2	1094.2	5539.9	6821.3	23,913.8 ± 350.80 c	17,092.5 ± 350.80 c
	N3W3	220.3	1094.2	5539.9	6854.4	28,885.0 ± 807.27 a	22,030.6 ± 807.27 a
Significance (F)	W	—	—	—	—	803.88 ***	795.63 ***
	N	—	—	—	—	325.68 ***	309.89 ***
	W × N	—	—	—	—	46.02 ***	46.02 ***
2022	N0W1	122.6	812.8	5369.1	6304.5	8756.3 ± 206.03 i	2451.8 ± 206.03 i
	N0W2	148.9	812.8	5369.1	6330.8	11,100.3 ± 313.70 gh	4769.5 ± 313.70 gh
	N0W3	175.2	812.8	5369.1	6357.1	16,093.9 ± 351.32 e	9736.8 ± 351.32 e
	N1W1	122.6	950.7	5369.1	6442.4	12,138.6 ± 163.83 fg	5696.2 ± 163.83 fg
	N1W2	148.9	950.7	5369.1	6468.7	20,392.4 ± 451.05 d	13,923.7 ± 451.05 d
	N1W3	175.2	950.7	5369.1	6495.0	23,508.1 ± 713.69 c	17,013.1 ± 713.69 c
	N2W1	122.6	996.6	5369.1	6488.3	12,552.5 ± 231.85 f	6064.2 ± 231.85 f
	N2W2	148.9	996.6	5369.1	6514.6	27,378.0 ± 753.37 b	20,863.4 ± 753.37 b
	N2W3	175.2	996.6	5369.1	6540.9	29,819.8 ± 629.38 a	23,278.9 ± 629.38 a
	N3W1	122.6	1042.6	5369.1	6534.3	10,707.0 ± 181.78 h	4172.7 ± 181.78 h
	N3W2	148.9	1042.6	5369.1	6560.6	23,721.3 ± 313.15 c	17,160.7 ± 313.15 c
	N3W3	175.2	1042.6	5369.1	6586.9	28,621.9 ± 245.20 ab	22,035.0 ± 245.20 ab
Significance (F)	W	—	—	—	—	1044.05 ***	1036.41 ***
	N	—	—	—	—	386.08 ***	371.57 ***
	W × N	—	—	—	—	52.59 ***	52.59 ***
2023	N0W1	143.2	682.1	5281.7	6107.0	5572.4 ± 170.51 h	−534.6 ± 170.51 h
	N0W2	173.9	682.1	5281.7	6137.7	10,524.8 ± 132.40 f	4387.1 ± 132.40 f
	N0W3	204.6	682.1	5281.7	6168.4	14,741.5 ± 225.44 e	8573.1 ± 225.44 e
	N1W1	143.2	827.6	5281.7	6252.5	9191.9 ± 133.61 g	2939.4 ± 133.61 g
	N1W2	173.9	827.6	5281.7	6283.2	18,675.9 ± 353.91 d	12,392.7 ± 353.91 d
	N1W3	204.6	827.6	5281.7	6313.9	22,340.7 ± 373.52 bc	16,026.8 ± 373.52 bc
	N2W1	143.2	876.1	5281.7	6301.0	9746.5 ± 281.15 fg	3445.5 ± 281.15 fg
	N2W2	173.9	876.1	5281.7	6331.7	22,722.3 ± 361.62 b	16,390.6 ± 361.62 b
	N2W3	204.6	876.1	5281.7	6362.4	26,080.3 ± 665.02 a	19,717.9 ± 665.02 a
	N3W1	143.2	924.6	5281.7	6349.5	9042.6 ± 205.69 g	2693.1 ± 205.69 g
	N3W2	173.9	924.6	5281.7	6380.2	21,482.4 ± 552.91 c	15,102.2 ± 552.91 c
	N3W3	204.6	924.6	5281.7	6410.9	25,288.3 ± 177.00 a	18,877.4 ± 177.00 a
Significance (F)	W	—	—	—	—	1710.28 ***	1696.08 ***
	N	—	—	—	—	442.83 ***	421.79 ***
	W × N	—	—	—	—	35.50 ***	35.51 ***

Note: Different letters following the same column of values indicate differences at the *p* < 0.05 level. *** indicates a significant difference at the *p* < 0.001 level, W indicates irrigation, and N indicates nitrogen fertilizer.

**Table 5 plants-15-00006-t005:** Optimized model combination schemes.

Targeted Economic Benefits (USD·ha^−1^)	2020		2021		2022		2023	
	Irrigation Amounts (mm)	Nitrogen Application Rates (kg·ha^−1^)	Irrigation Amounts (mm)	Nitrogen Application Rates (kg·ha^−1^)	Irrigation Amounts (mm)	Nitrogen Application Rates (kg·ha^−1^)	Irrigation Amounts (mm)	Nitrogen Application Rates (kg·ha^−1^)
19,000–17,000	416.5–465.3	146.0–204.5	295.5–323.9	127.1–253.1	270.5–314.3	91.5–259.3	379.4–401.0	139.2–256.8
21,000–19,000	503.1–534.9	219.8–259.0	332.4–367.9	117.1–235.7	300.7–335.0	127.2–253.1	422.3–453.5	140.2–206.6
23,000–21,000	—	—	365.6–459.3	120.9–234.5	340.2–381.1	124.3–235.1	589.4–620.8	127.4–236.1
25,000–23,000	—	—	441.6–518.9	110.6–193.7	387.1–463.7	112.6–216.5	477.3–540.4	123.4–222.9
27,000–25,000	—	—	470.4–536.2	152.6–225.8	430.5–491.9	138.4–222.4	542.0–583.7	161.9–224.0
29,000–27,000	—	—	529.6–572.1	203.1–257.7	469.9–511.4	194.8–254.0	609.4–640.3	226.4–275.2

**Table 6 plants-15-00006-t006:** Crop coefficients under different onion growth stages from 2021 to 2023.

Growth Stages	Seedling Stage	Leaf Development Stage	Bulbification Stage	Maturity Stage
K_c_	0.7	0.7	1.05	0.75

**Table 7 plants-15-00006-t007:** Coded values of experimental factors and the experimental design.

Treatment	Irrigation Amount (mm)				Irrigation Code Value X_1_	Nitrogen Application Rate (kg·ha^−1^)	Nitrogen Application Rate Code Value X_2_
	2020 Year	2021 Year	2022 Year	2023 Year			
N0W1	408.31	447.87	400.20	462.24	—	0	—
N0W2	495.81	543.85	485.96	561.29	—	0	—
N0W3	583.30	639.82	571.72	660.34	—	0	—
N1W1	408.31	447.87	400.20	462.24	1	198	1
N1W2	495.81	543.85	485.96	561.29	0	198	1
N1W3	583.30	639.82	571.72	660.34	−1	198	1
N2W1	408.31	447.87	400.20	462.24	1	264	0
N2W2	495.81	543.85	485.96	561.29	0	264	0
N2W3	583.30	639.82	571.72	660.34	−1	264	0
N3W1	408.31	447.87	400.20	462.24	1	330	−1
N3W2	495.81	543.85	485.96	561.29	0	330	−1
N3W3	583.30	639.82	571.72	660.34	−1	330	−1

## Data Availability

The original contributions presented in the study are included in the article; further inquiries can be directed to the corresponding author.

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
