# Peer review of "Optimizing Water and Nitrogen Management Strategies to Unlock the Production Potential for Onion in the Hexi Corridor of China: Insights from Economic Analysis"

_plants, 2025, doi:10.3390/plants15010006_

Round 1

Reviewer 1 Report

Comments and Suggestions for Authors

This manuscript presents a comprehensive four-year field experiment investigating the effects of water-nitrogen coupling on onion yield, quality, water-nitrogen use efficiency, and economic benefits under mulched drip irrigation in the Hexi Corridor. The study is data-rich, but not well-structured and regionally relevant.  And, several issues related to data presentation, statistical clarity, and scientific novelty need to be addressed.
Key issues:
-The authors conducted a complete 3 × 3 water–nitrogen factorial experiment, yet they coded the factors as 0, 0.5, 1. This coding scheme is appropriate for orthogonal designs, not for a complete factorial layout, and its use here is therefore incorrect.
-In Figure 2 only air temperature, ETâ‚€ and precipitation are shown; irrigation amounts are not provided.
-A separate regression equation was fitted for each individual year, which greatly limits the generality of the results.
-The regression model is useful, but its predictive accuracy and transferability to other years or sites are not validated. Consider:
Including a validation year or independent dataset.
-Discussing limitations of the model under climate variability or soil heterogeneity.
Other comments:
-The interaction terms (W×N) are significant in some years but not others. Please discuss the implications of this inconsistency.
-Post-hoc tests (e.g., LSD) are mentioned but not fully described. Clarify the multiple comparison procedure used and whether p-values were adjusted for family-wise error rates.

-The economic model assumes fixed input prices and output prices. Please discuss how price volatility (e.g., fertilizer, water, onion market price) might affect the optimal input recommendations.
-Labor costs are not explicitly mentioned. If excluded, please justify.

-While the title mentions “sustainable production,” the paper focuses heavily on economic benefits. Consider:
-Including environmental indicators such as N leaching, Nâ‚‚O emissions, or carbon footprint.
-Discussing trade-offs between economic and environmental objectives.

-The phrase “water-nitrogen-economic benefit coupling model” is repeated multiple times. Consider rephrasing for clarity and brevity.
-Please quantify key findings more precisely (e.g., “economic benefits increased by 25.3%” instead of “significantly increased”).
-Figure 3 and 4: Axis labels are small and hard to read. Please increase font size.
-Table 6: Consider moving to Supplementary Materials due to its size, or summarize key trends in the main text.
-Table 7: The confidence intervals are helpful but lack units in some rows. Please add units for clarity.
-Language and Style
The manuscript contains numerous grammatical errors and awkward phrasings. A thorough language editing by a native speaker or professional service is strongly recommended.
Examples:
“the yield-increasing effect is significantly diminishing” → “the yield-increasing effect diminishes significantly”
“introducing inappropriate” → “inappropriate introduction”

Author Response

Response to Reviewers 

Reply to Reviewer #1: This manuscript presents a comprehensive four-year field experiment investigating the effects of water-nitrogen coupling on onion yield, quality, water-nitrogen use efficiency, and economic benefits under mulched drip irrigation in the Hexi Corridor. The study is data-rich, but not well-structured and regionally relevant. And, several issues related to data presentation, statistical clarity, and scientific novelty need to be addressed.

Dear Reviewer,

We feel great thanks for your professional review work on our article. According to your constructive suggestions, we have made extensive corrections to our previous draft, the detailed corrections are listed below.

Responds to reviewer’s comments:

Comment 1:The authors conducted a complete 3 × 3 water–nitrogen factorial experiment, yet they coded the factors as 0, 0.5, 1. This coding scheme is appropriate for orthogonal designs, not for a complete factorial layout, and its use here is therefore incorrect.

Response 1: Your suggestion really means a lot to us. According to your request, we have modified the factor encoding to -1,0,1, at the same time, we have modified the relevant formulas and data in the article, please refer to the attached revised manuscript for details of the revisions.

Comment 2:In Figure 2 only air temperature, ETâ‚€ and precipitation are shown; irrigation amounts are not provided.

Response 2: Thanks for your suggestion, we have increased the figure of irrigation amounts and nitrogen application amounts (Figure 3).

Comment 3:A separate regression equation was fitted for each individual year, which greatly limits the generality of the results.

Response 3: Thank you for your reminder. We have fitted the regression equation based on the average data from 2020 to 2022,the predictive accuracy of the regression model was validated using the 2023 dataset as an independent validation set. The results showed a strong agreement between predicted and observed economic benefits. (L412-426)

Comment 4:The regression model is useful, but its predictive accuracy and transferability to other years or sites are not validated. Consider:Including a validation year or independent dataset.

Response 4: Thank you for your reminder. We have fitted the regression equation based on the average data from 2020 to 2022,the predictive accuracy of the regression model was validated using the 2023 dataset as an independent validation set. The results showed a strong agreement between predicted and observed economic benefits (MRE = 9.53%, R2 = 0.988, RMSE = 1755.92 USD·ha-1), confirming the robustness of model within the experimental conditions. (L412-426) 

Comment 5:Discussing limitations of the model under climate variability or soil heterogeneity.

Response 5: Thanks for your suggestion. We have discussed limitations of the model under climate variability or soil heterogeneity.(L427-429) 

Comment 6:The interaction terms (W×N) are significant in some years but not others. Please discuss the implications of this inconsistency.

Response 6: Thanks for your suggestion. We have discussed the differences in the influence of water-single interaction in different years.(L668-672) 

Comment 7:Post-hoc tests (e.g., LSD) are mentioned but not fully described. Clarify the multiple comparison procedure used and whether p-values were adjusted for family-wise error rates.

Response 7: Thank you for pointing this out. We used the LSD multiple comparison method in SPSS 22.0 software, no adjustment for family-wise error rates was applied.

Comment 8:The economic model assumes fixed input prices and output prices. Please discuss how price volatility (e.g., fertilizer, water, onion market price) might affect the optimal input recommendations.

Response 8:Thank you for pointing this out. We calculate the economic benefits of Onions based on the actual market price of that year and do not assume that the input price and output price are fixed.

Comment 9:Labor costs are not explicitly mentioned. If excluded, please justify.

Response 9: Thank you for pointing this out. Labor costs were included in the ‘Other inputs’ category.

Comment 10:While the title mentions “sustainable production,” the paper focuses heavily on economic benefits. Consider:Including environmental indicators such as N leaching, N2O emissions, or carbon footprint. Discussing trade-offs between economic and environmental objectives.

Response 10:Thanks for your suggestion. We revised the title of the paper to make it more relevant to the content. In future research, we will consider environmental indicators such as N leaching, N2O emissions, or carbon footprint. 

Comment 11:The phrase “water-nitrogen-economic benefit coupling model” is repeated multiple times. Consider rephrasing for clarity and brevity.

Response 11:Thank you very much for your comments and suggestions. We have uniformly replaced “water-nitrogen-economic benefit coupling model” with “water-nitrogen-economic model” or “regression model” throughout the article.

Comment 12:Please quantify key findings more precisely (e.g., “economic benefits increased by 25.3%” instead of “significantly increased”).

Response 12:Thanks for your suggestion. We have precisely expressed the relevant statements in the article.

Comment 13:Figure 3 and 4: Axis labels are small and hard to read. Please increase font size.

Response 13:Thanks for your suggestion. We redrew Figure 3 and 4, increased the font size of axis labels.

Comment 14:Table 6: Consider moving to Supplementary Materials due to its size, or summarize key trends in the main text.

Response 14:Thank you for pointing this out. We have summarize key trends of table 6 in the main text.

Comment 15:Table 7: The confidence intervals are helpful but lack units in some rows. Please add units for clarity.

Response 15:We were really sorry for our careless mistakes. Thank you for your reminder. We have added units.  

Comment 16:The manuscript contains numerous grammatical errors and awkward phrasings. A thorough language editing by a native speaker or professional service is strongly recommended. “the yield-increasing effect is significantly diminishing” → “the yield-increasing effect diminishes significantly”“introducing inappropriate” → “inappropriate introduction”

Response 16:Thank you very much for your comments and suggestions. We have revised strictly in accordance with the comments, combed the full text, checked sentence structure and English grammar, and made professional linguistic touch-ups.

Please refer to the attached revised manuscript for details of the revisions.

Sincerely,

The Authors

Reviewer 2 Report

Comments and Suggestions for Authors

Comments and Suggestions for Authors

Title: Optimizing Water and Nitrogen for Sustainable Onion Production in the
Hexi Corridor of China: Insights into Economic Analysis

Dear Authors and Editors

The research results presented in the manuscript fall within the publishing profile of the journal Plants. The research topic is original and relevant to the field of agricultural sciences and the discipline of horticulture.

The title of the manuscript is interesting, and the issues addressed are relevant in conditions of water shortage and balanced fertilization. A major advantage of the article submitted for review is the results obtained on the basis of four years of field research. Please dispel my doubts. Why was it decided to use high doses of nitrogen in the study? After all, onions are a vegetable intended for human consumption. What are the permissible doses of N in vegetable fertilization in the country where the study was conducted? High yields may not be the most important thing, but the quality of the crop is.

The authors presented a large number of results in successive, large tables, but their descriptions are very repetitive. The discussion section is well-written. The conclusions should be revised according to the comments below. References were used appropriately.

In order to increase the usefulness of the article, Authors must refer to the following points. Additions should be made to increase the scientific value of the manuscript.

Remarks

  1. Materials and Methods: Subsection 2.1. Please provide soil classification according to IUSS Working Group 2014, 2022. Section 2.2. What does it mean that N rates were applied according to local tradition? After all, in both scientific research and agricultural practice, N rates are determined according to specific criteria. Phosphorus and potassium rates should be reported in kg P ha-1 and kg K ha-1. Subsubsection 2.3.2. Please add the names of the apparatuses (city, country of production) used during chemical analyses. Line 207 - Are the N doses in kg kg-1?
  2. Results and analysis: Table 3, 5, and 6 -To improve the readability of the numerical data, please consider a different format for presenting the research results. I suggest presenting the analysis of subsequent parameters in separate tables. Table 4 - Yields should be presented in t ha-1. Table 6 - You should consider changing your local unit to an international unit.
  3. Conclusions: Lines 702 and 703 - Instead of the N2W3 and N2W2 designations, please provide nitrogen rates and irrigation amounts. Line 708 ...over- and under-irrigation... So what kind of irrigation? Directions for further research should be added.

Specific remarks

  1. Reference no. 30 - not cited in the manuscript text.
  2. References: It should be adapted to publishing requirements.

Best regards

Author Response

Response to Reviewers 

Reply to Reviewer #2: The research results presented in the manuscript fall within the publishing profile of the journal Plants. The research topic is original and relevant to the field of agricultural sciences and the discipline of horticulture. The title of the manuscript is interesting, and the issues addressed are relevant in conditions of water shortage and balanced fertilization. A major advantage of the article submitted for review is the results obtained on the basis of four years of field research. Please dispel my doubts. Why was it decided to use high doses of nitrogen in the study? After all, onions are a vegetable intended for human consumption. What are the permissible doses of N in vegetable fertilization in the country where the study was conducted? High yields may not be the most important thing, but the quality of the crop is. The authors presented a large number of results in successive, large tables, but their descriptions are very repetitive. The discussion section is well-written. The conclusions should be revised according to the comments below. References were used appropriately. In order to increase the usefulness of the article, Authors must refer to the following points. Additions should be made to increase the scientific value of the manuscript.

Dear Reviewer,

We feel great thanks for your professional review work on our article. According to your constructive suggestions, we have made extensive corrections to our previous draft, the detailed corrections are listed below.

Responds to reviewer’s comments:

Comment 1:Why was it decided to use high doses of nitrogen in the study? After all, onions are a vegetable intended for human consumption. What are the permissible doses of N in vegetable fertilization in the country where the study was conducted? High yields may not be the most important thing, but the quality of the crop is.

Response 1: Thank you for pointing this out. The low (N1), medium (N2), and high (N3) nitrogen dosage levels were set at 60%, 80%, and 100%, respectively, of the local recommended fertilization level of 330 kg ha-1. We agree with your viewpoint that while achieving high crop yields, crop quality must also be guaranteed.

Comment 2:Materials and Methods: Subsection 2.1. Please provide soil classification according to IUSS Working Group 2014, 2022.

Response 2: In accordance with your comment, we have added the soil classification. (L131)

Comment 3:Section 2.2. What does it mean that N rates were applied according to local tradition? After all, in both scientific research and agricultural practice, N rates are determined according to specific criteria. Phosphorus and potassium rates should be reported in kg P ha-1 and kg K ha-1.

Response 3: Thank you for pointing this out. Our initial statement in the article might have been incorrect. The traditional nitrogen application rate refers to the recommended rate provided by the local agricultural extension department. Consequently, we have revised the relevant description in the article. Additionally, the units for phosphorus and potassium fertilizer application rates have been corrected.

Comment 4:Subsubsection 2.3.2. Please add the names of the apparatuses (city, country of production) used during chemical analyses.

Response 4: Thank you for your suggestion. We have added the names and origins of the apparatuses. (L180-183) 

Comment 5:Line 207 - Are the N doses in kg kg-1?

Response 5: We sincerely apologize for this careless mistake. Thank you for reminding us. We have corrected the unit for nitrogen. (L216) 

Comment 6:Results and analysis: Table 3, 5, and 6 -To improve the readability of the numerical data, please consider a different format for presenting the research results. I suggest presenting the analysis of subsequent parameters in separate tables.

Response 6: Thanks for your suggestion. We analyzed each table separately for table 3, 5, and 6.

Comment 7:Table 4 - Yields should be presented in t ha-1.

Response 7: In accordance with your comment, we have changed the yield units to t ha-1. (L315-319)  

Comment 8:Table 6 - You should consider changing your local unit to an international unit.

Response 8: In accordance with your comment, we have converted the local unit to international unit.

Comment 9:Conclusions: Lines 702 and 703 - Instead of the N2W3 and N2W2 designations, please provide nitrogen rates and irrigation amounts.

Response 9: Thanks for your suggestion, we supplemented the nitrogen rates and irrigation amounts of N2W3 and N2W2 treatments in the conclusion. (L766-768) 

Comment 10:Line 708 ...over- and under-irrigation... So what kind of irrigation? Directions for further research should be added.

Response 10: Thank you very much for your comments and suggestions. We have removed the incorrect expressions and added directions for further research. (L778-780) 

Comment 11:Reference no. 30 - not cited in the manuscript text.

Response 11: We were really sorry for our careless mistakes. Thank you for your reminder.  We have revised. (L597) 

Comment 12:References: It should be adapted to publishing requirements.

Response 12: Thanks for your suggestion, we have revised the format of the references in accordance with the requirements of the journal.

Please refer to the attached revised manuscript for details of the revisions.

Sincerely,

The Authors

Round 2

Reviewer 1 Report

Comments and Suggestions for Authors

The manuscript has been significantly improved.

I am satisfied with the author’s revisions and have no further suggestions.

I recommend that it be published as soon as possible.

Author Response

Reply to Reviewer #1: The manuscript has been significantly improved. I am satisfied with the author’s revisions and have no further suggestions. I recommend that it be published as soon as possible.

Dear Reviewer,

We are pleased to hear that you are satisfied with the revisions and that you recommend publication. We appreciate your time and valuable input throughout the review process.  Thank you very much for your comments and suggestions. We rechecked the sentence structure and English grammar to ensure they met the requirements of the journal.

Please refer to the attached revised manuscript for details of the revisions.

Sincerely,

The Authors
